# Nuclear and cytosolic J-domain proteins provide synergistic control of Hsf1 at distinct phases of the heat shock response

Carmen Ruger-Herreros[1,2†], Lucia Svoboda[1†], Gurranna Male[3†], Aseem Shrivastava[1], Markus Höpfler[4], Katharina Jetzinger[1], Jiří Koubek[1], Günter Kramer[1], Fabian den Brave[5*], Axel Mogk[1*], David S Gross[3*], Bernd Bukau[1*]

[1]Center for Molecular Biology of Heidelberg University (ZMBH), DKFZ-ZMBH Alliance, Heidelberg, Germany; [2]Instituto de Biomedicina de Sevilla (IBiS), Hospital Universitario Virgen del Rocío/CSIC/Universidad de Sevilla, Sevilla, Spain; [3]Department of Biochemistry and Molecular Biology, Louisiana State University Health Sciences Center, Shreveport, United States; [4]Centre for Genomic Regulation (CRG), Genome Biology Programme, Barcelona, Spain; [5]Institut für Biochemie und Molekularbiologie, Universität Bonn, Bonn, Germany

**\*For correspondence:**
fden@uni-bonn.de (FdB);
a.mogk@zmbh.uni-heidelberg.
de (AM);
david.gross@lsuhs.edu (DSG);
bukau@zmbh.uni-heidelberg.
de (BB)

†These authors contributed
equally to this work

**Reviewing Editor:** Luke
Wiseman, Scripps Research
Institute, United States

## eLife Assessment

This **valuable** study focuses on defining how the HSP70 chaperone system utilizes J-domain proteins to regulate the heat shock response-associated transcription factor HSF1. Using a combination of orthogonal techniques in yeast, this article provides **compelling** evidence that the J-domain protein Apj1 facilitates attenuation of HSF1 transcriptional activity through a mechanism involving its dissociation from heat shock gene promoter regions. This work generates new insight into the mechanism of HSF1 transcriptional regulation and is a significant contribution of broad interest to cell biologists interested in proteostasis, chaperone networks, and stress-responsive signaling.

**Abstract** The heat shock response (HSR) is the major defense mechanism against proteotoxic stress in the cytosol and nucleus of eukaryotic cells. Initiation and attenuation of the response are mediated by stress-dependent regulation of heat shock transcription factors (HSFs). *Saccharomyces cerevisiae* encodes a single HSF (Hsf1), facilitating the analysis of HSR regulation. Hsf1 is repressed by Hsp70 chaperones under non-stress conditions and becomes activated under proteotoxic stress, directly linking protein damage and its repair to the HSR. J-domain proteins (JDPs) are essential for targeting of Hsp70s to their substrates, yet the specific JDP(s) regulating Hsf1 and connecting protein damage to HSR activation remain unclear. Here, we show that the yeast nuclear JDP Apj1 primarily controls the attenuation phase of the HSR by promoting Hsf1's displacement from heat shock elements in target DNA. In *apj1Δ* cells, HSR attenuation is significantly impaired. Additionally, yeast cells lacking both Apj1 and the major JDP Ydj1 exhibit increased HSR activation even in non-stress conditions, indicating their distinct regulatory roles. Apj1's role in both nuclear protein quality control and Hsf1 regulation underscores its role in directly linking nuclear proteostasis to HSR regulation. Together, these findings establish the nucleus as key stress-sensing signaling hub.

## Introduction

The cellular protein quality control (PQC) system employs various strategies, including protein folding, degradation, sequestration, and disaggregation, which are primarily executed by molecular chaperones and components of the ubiquitin-proteasome and autophagy systems. To maintain proteostasis even under proteotoxic stress, cells must enhance PQC capacity by increasing the expression of its components in a cell compartment-specific manner (*Pincus, 2020*; *Shpilka and Haynes, 2018*; *Walter and Ron, 2011*). For the cytosolic and nuclear PQC system, expression is principally regulated by the evolutionarily conserved heat shock transcription factor, Hsf1 (*Kmiecik and Mayer, 2022*). Aging can reduce Hsf1 activity, and by extension the PQC capacity, which in humans is linked to neurodegenerative diseases including Alzheimer's and Parkinson's (*Li et al., 2017*). Conversely, high HSF1 activity can promote the proliferation and spread of certain forms of cancer (*Pessa et al., 2024*). Understanding HSF1 activity is therefore critical for therapeutic strategies.

In *Saccharomyces cerevisiae*, Hsf1 drives the transcription of genes encoding PQC components during proteotoxic stress (*Pincus, 2020*), controlling the expression of 46 target genes (*Pincus et al., 2018*). Hsf1 becomes activated upon heat shock and binds to heat shock elements (HSEs) in the upstream activating regions of heat shock genes to initiate a transient response, which later is downregulated in a negative feedback loop. Hsf1 is essential for yeast growth, maintaining basal levels of Hsp70 and Hsp90 chaperones that support protein homeostasis (*Solís et al., 2016*). However, overactivation can impair growth of yeast cells (*Halladay and Craig, 1995*), which underscores the biological importance of a tunable Hsf1 activity.

Hsp70 chaperones are central to regulating Hsf1 activity across species (*Kmiecik and Mayer, 2022*). Hsf1's activity is furthermore modulated by phosphorylation, allowing it to integrate signals from diverse pathways (*Zheng et al., 2016*). Hsf1 phosphorylation increases during heat shock but is not essential for its basal activity (*Zheng et al., 2016*). Hsp70 binds and represses Hsf1 activity, keeping it inactive (*Peffer et al., 2019*; *Zheng et al., 2016*). Accordingly, Hsf1 is more active in yeast lacking the two major Hsp70 proteins Ssa1 and Ssa2 (*Halladay and Craig, 1995*). Hsp70 influences both the initiation and attenuation phases of the heat shock response (*Garde et al., 2023*; *Krakowiak et al., 2018*) through a titratable repressor function, allowing regulation based on misfolded protein levels (*Masser et al., 2020*; *Pincus, 2020*).

Hsp70 chaperones rely on J-domain proteins (JDPs) to bind specific substrates (*Kampinga and Craig, 2010*; *Rosenzweig et al., 2019*; *Ruger-Herreros et al., 2024*). JDPs preselect substrates and transfer them to Hsp70, facilitating ATP hydrolysis-driven trapping in its binding pocket (*Rosenzweig et al., 2019*). In yeast, the nuclear-cytosolic JDP, Sis1, targets Hsf1 to Ssa1 (*Masser et al., 2019*) and plays a dual role in PQC and Hsf1 regulation, linking protein damage accumulation and heat shock response (HSR) activation. Sis1 relocalizes to PQC foci on the endoplasmic reticulum (ER) and nucleus during stress (*Feder et al., 2021*) and binds misfolded proteins at nuclear inclusions (INQ) via interaction with the nuclear sequestrase Btn2 (*Ho et al., 2019*). It also binds orphan ribosomal proteins (oRPs) in the nucleolus under stress, preventing the formation of solid aggregates (*Ali et al., 2023*). As Sis1 targets diverse PQC sites, its substrate pool changes, reducing its ability to repress Hsf1, thereby triggering Hsf1 activation and initiation of the HSR. Accordingly, impairing ribosome biogenesis activates Hsf1 (*Albert et al., 2019*; *Tye et al., 2019*).

Interestingly, Sis1 depletion or a full gene deletion (viable upon overexpression of Tti1) induces only a partial HSR, while deletion of *SSA1* and *SSA2* fully activates Hsf1 (*Schilke and Craig, 2022*), suggesting other JDPs are involved. Additionally, Sis1 is dispensable for the HSR attenuation phase, which depends on Hsp70 (*Garde et al., 2023*; *Krakowiak et al., 2018*), indicating the role of an unidentified JDP co-chaperone.

Here, we aimed to identify the JDP that represses Hsf1 during the attenuation phase. We focused on the nuclear JDP Apj1, which increases during heat shock and targets proteins deposited at INQ for proteasomal degradation (*den Brave et al., 2020*; *Sahi et al., 2013*). Its nuclear localization and PQC function make Apj1 an attractive candidate for linking the status of nuclear PQC to Hsf1 activity control. We show that Apj1 represses Hsf1 activity during the attenuation phase by promoting Hsf1 displacement from HSEs. Its role is further highlighted by the full heat shock response observed in cells lacking Apj1 and Ydj1, the major class A JDP of yeast, under non-stress conditions. The dual role of Apj1 in nuclear PQC and HSR regulation indicates that the nucleus is key to linking proteostasis with Hsf1 regulation.

## Results

### Apj1 represses Hsf1 activity during the attenuation phase of the heat shock response

To study the role of yeast JDPs in regulating Hsf1 activity, we focused on Apj1 due to its nuclear localization and involvement in nuclear PQC (*den Brave et al., 2020*). Apj1, a class A JDP, originated from the major JDP Ydj1 through gene duplication and diversification (*Sahi et al., 2013*). To explore Apj1's role in HSR regulation, we monitored the expression of the nuclear sequestrase Btn2 in wild-type (wt) and *apj1Δ* cells during heat shock (30–38°C), as Btn2 expression strictly depends on Hsf1 (*Pincus et al., 2018*; *Solís et al., 2016*). Btn2 only transiently accumulates during heat shock owing to its rapid turnover by proteasomal degradation (*Malinovska et al., 2012*; *Miller et al., 2015b*). The decline of Btn2 levels at later timepoints of heat shock thus reports on the attenuation of the stress response. We observed a strong but transient increase in Btn2 levels in heat-shocked (from 30°C to 38°C) wt cells (*Figure 1A*), while Btn2 levels remained high for up to 120 min post temperature upshift in *apj1Δ* cells. To investigate whether the persistent accumulation of Btn2 in *apj1Δ* cells results from stabilization of Btn2, we added cycloheximide (CHX) 10 min post heat shock to inhibit protein synthesis. Here, Btn2 levels rapidly vanished in wt and *apj1Δ* cells (*Figure 1B*), indicating that the high Btn2 levels in *apj1Δ* cells rely on ongoing transcription and translation. Similar results were obtained with a GFP reporter gene placed under control of the *BTN2* promoter (*Figure 1—figure supplement 1A and B*). The GFP reporter protein was made unstable by fusion to the N-terminal domain of Btn2. *apj1Δ* but not wt cells showed persistent GFP accumulation that was dependent on continuous protein synthesis as it was no longer observed after CHX addition. These findings indicate a role of Apj1 in the attenuation of the heat shock response.

Recently, Apj1 was identified as a negative regulator of Hsf1 through a genetic screen for feedback regulators of the yeast HSR (*Garde et al., 2024*). This screen used a YFP reporter gene under control of a synthetic Hsf1 promoter. However, substantial differences between the expression of this synthetic Hsf1 reporter and authentic Hsf1 targets have been observed (*Schilke and Craig, 2022*). To provide direct evidence for a global regulatory function of Apj1 in HSR attenuation, we performed ribosome profiling in wt and *apj1Δ* cells. Previous studies indicated that changes in gene expression during heat shock in yeast cells can be faithfully tracked using translatome data (*Mühlhofer et al., 2019*). We analyzed translatomes of non-stressed cells (0 min), shortly after heat shock (10 min), and after prolonged heat shock (60 min) to assess HSR attenuation. To compare the regulatory impact of Apj1 and Sis1, we included *sis1-4xcga* cells. In these cells, four CGA codons are fused to the end of the *SIS1* coding sequence which activates ribosome quality control (RQC) pathways leading to degradation of the Sis1 de novo synthesized chains and strongly depleted Sis1 levels (14.9 ± 3%) (*Sitron and Brandman, 2020*). We also analyzed *apj1Δ sis1-4xcga* cells for overlapping JDP functions. Upregulation of Hsf1 target genes at 10 min of heat shock was similar across all strains (*Figure 1C and D*, *Figure 1—figure supplement 1C*). However, *apj1Δ* cells exhibited increased expression of Hsf1 targets at 60 min (*Figure 1C and D*, *Figure 1—figure supplement 1C/D*), indicating a general defect in HSR attenuation and underlining Apj1's central role in re-inactivating Hsf1. The effect of *sis1-4xcga* on HSR attenuation varied, with some genes showing delayed repression (e.g., *HSP104*) and others unaffected (e.g., *BTN2*) (*Figure 1C*). No significant differences in Hsf1 target gene expression were found between wt and *sis1-4xcga* cells at 60 min, while expression was significantly increased in *apj1Δ* cells (p=3.724e-05) (*Figure 1D*, *Figure 1—figure supplement 1D*). Notably, there was no further deregulation in *apj1Δ sis1-4xcga* compared to *apj1Δ* cells (*Figure 1C and D*, *Figure 1—figure supplement 1C and D*), underscoring Apj1's specific function in HSR attenuation.

We next monitored the localization of Hsf1-GFP in wt and *apj1Δ* cells to assess Hsf1 activity. Hsf1-GFP predominantly localized to the nucleus, showing diffuse staining under non-stress conditions (*Figure 2A*), in agreement with previous findings (*Chowdhary et al., 2019*). Upon heat shock, Hsf1-GFP formed discrete nuclear foci, indicating its binding to clusters of coalesced heat shock genes from distant chromosomal locations (*Chowdhary et al., 2019*). In wt cells, the number and intensity of Hsf1-GFP foci declined during prolonged heat shock, reflecting the disassembly of heat shock gene condensates and Hsf1 inactivation (*Figure 2A*; *Chowdhary et al., 2022*). In contrast, *apj1Δ* cells maintained Hsf1-GFP foci throughout the 60 min of heat shock (*Figure 2A*), indicating persistent Hsf1 activation.

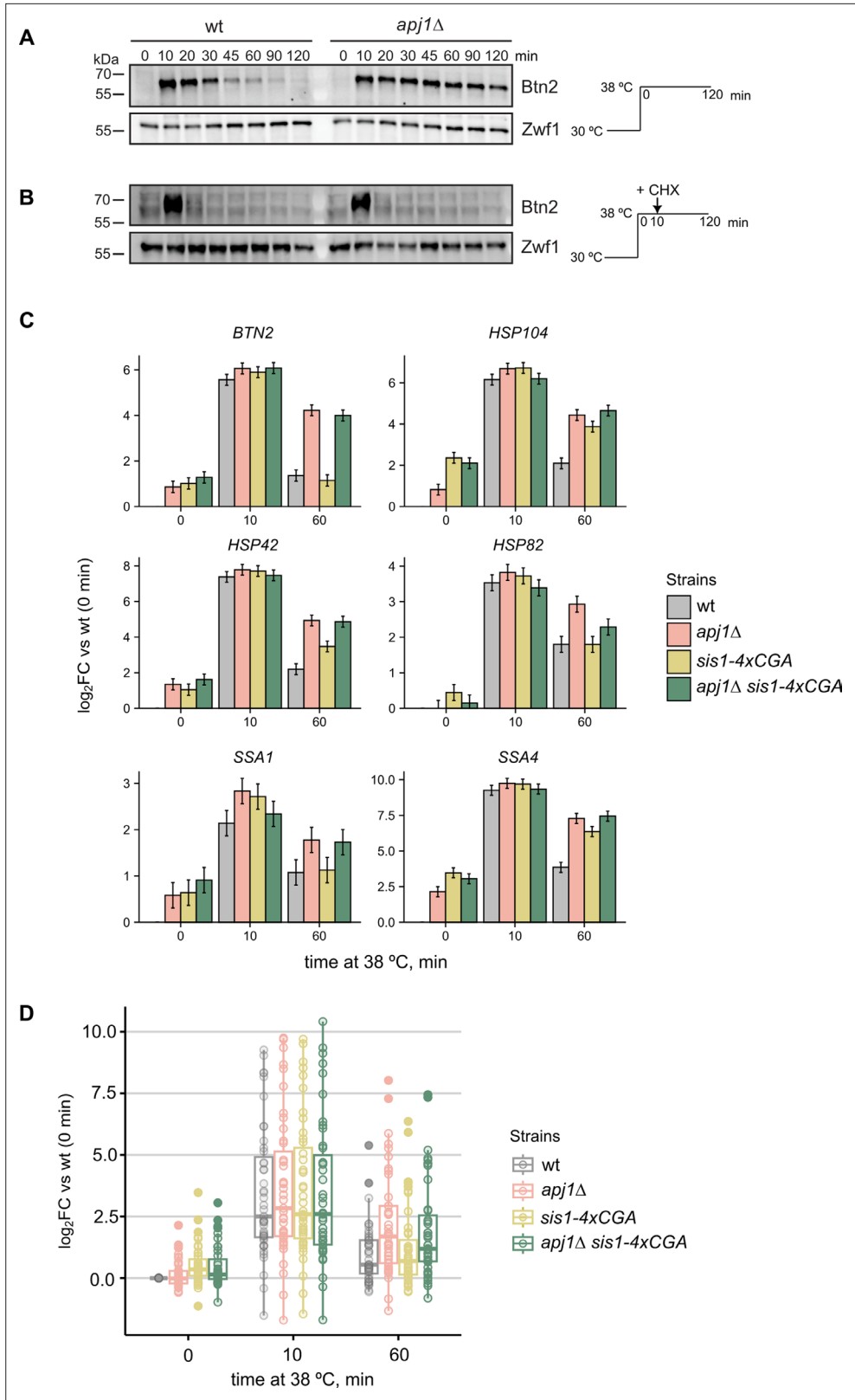

**Figure 1.** Attenuation of the Hsf1-mediated heat shock response is defective in *S. cerevisiae apj1Δ* cells. (**A, B**) *S. cerevisiae* wt and *apj1Δ* cells were grown at 30°C till logarithmic growth phase and shifted to 38°C. After 10 min, cycloheximide (CHX) was added to inhibit protein synthesis (**B**). Total cell extracts were prepared and levels of the heat shock protein Btn2 were determined by western blot analysis at the indicated time points after heat shock.

*Figure 1 continued on next page*

*Figure 1 continued*

Zwf1 levels were determined as loading control. (**C**) Changes in expression levels of selected Hsf1-target genes were determined by ribosome profiling in indicated yeast strains grown at 30°C and subjected to heat shock at 38°C for 0, 10, or 60 min. Levels of translated mRNAs were normalized to respective levels determined in wild-type cells prior to heat shock and shown in a log2-fold change-scale (log2FC) with the standard error (n=2). (**D**) Changes in expression levels of all 46 Hsf1 targets were determined and normalized to respective levels determined in wild-type cells prior to heat shock. Significance was determined by Wilcox test (**p<0.01; ***p<0.001).

The online version of this article includes the following source data and figure supplement(s) for figure 1:

**Source data 1.** Original western blots with bands shown in *Figure 1* highlighted.

**Source data 2.** Original western blots of *Figure 1*.

**Figure supplement 1.** Attenuation of the Hsf1-mediated heat shock response is defective in *S. cerevisiae apj1Δ* cells.

**Figure supplement 1—source data 1.** Original western blots with bands shown in *Figure 1—figure supplement 1* highlighted.

**Figure supplement 1—source data 2.** Original western blots shown in *Figure 1—figure supplement 1*.

**Figure supplement 2.** Expression levels of yeast J-domain proteins (JDPs).

---

The coalescence of heat shock genes depends on activated Hsf1 (*Chowdhary et al., 2019*) and is used here as an additional microscopic assay of Hsf1 activity. We employed a yeast strain expressing GFP-LacI with varying numbers of *lacO* binding cassettes linked to *HSP104* and *HSP12*, located on chromosomes XII and VI, respectively (*Chowdhary et al., 2017*). The position of the two heat shock genes can be visualized as large (*HSP104*) and small (*HSP12*) fluorescent foci. The presence of a single focus indicates colocalization and coalescence of both genes. Heat shock strongly increased *HSP104* and *HSP12* gene coalescence in wt cells from 24.7% (before heat shock) to 64.7% (10 min post heat shock) (*Figure 2B*). However, coalescence levels dropped back to pre-stress levels afterwards. In *apj1Δ* cells, the percentage of cells showing gene coalescence before heat shock was higher (39%) and increased to 57,3% post heat stress, similar to wt cells. Notably, we did not observe dissociation of the coalesced *HSP104* and *HSP12* genes at later time points, with coalescence remaining around 60% (*Figure 2B*). These findings demonstrate the absence of HSR attenuation in *apj1Δ* cells due to persistent Hsf1 activation.

## Apj1 occupancy of HSEs anti-correlates with that of Hsf1

To explore Apj1's impact on Hsf1, we measured Hsf1 levels and phosphorylation in wt and *apj1Δ* cells during heat shock. Heat shock triggers Hsf1 phosphorylation, enhancing its activity (*Zheng et al., 2016*). We observed similar transient Hsf1 phosphorylation in both cell types, indicated by the reduced mobility of phosphorylated Hsf1 in SDS-gels (*Figure 3—figure supplement 1A*), and no differences in Hsf1 levels were noted, excluding altered stability or phosphorylation status as reasons for the increased Hsf1 activity in *apj1Δ* cells during the attenuation phase.

We next tested whether Apj1 interacts with active, HSE-bound Hsf1 using chromatin immunoprecipitation (ChIP) after a 30 min heat shock at 42°C using GFP-Apj1. We used the GFP-Apj1-[34]AAA[37] variant (GFP-Apj1*) with a mutated [34]HPD[37] motif in the J-domain that is essential for Hsp70 interaction, hypothesizing that the block of substrate transfer to Hsp70 would stabilize GFP-Apj1*-Hsf1 interactions. GFP-Apj1* specifically bound to promoter regions of several Hsf1-controlled heat shock genes (e.g., *SSA4*, *HSP42*, *UBI4 BTN2*) (*Figure 3A /B*), while no binding was observed in GFP-only ChIP controls (*Figure 3A/B*) or for non-heat shock genes like *TOS1* (*Figure 3B*). Motif analysis (MEME suite, *Bailey et al., 2015*) of the CHIP peaks revealed a conserved sequence present in all top 18 peaks that is highly similar to the canonical HSE (*Figure 3C*). This indicates Apj1's specific pull-down with HSE-harboring DNA fragments, presumably via interaction with Hsf1.

Next, we assessed the dynamics of the Hsf1 and Apj1 interaction with upstream activating sequences (UASs) of Hsf1-controlled heat shock genes. We used Apj1-Myc$_{13}$ harboring an intact HPD motif which ensures dynamic interactions between Apj1 and Hsf1 and allows us to determine by qPCR the temporal changes of UAS occupancies during heat shock. Heat shock strongly increased Hsf1 occupancy at UASs of heat shock genes within 5 min (*Figure 3D*, *Figure 3—figure supplement 1B*), which then declined, reflecting the attenuation phase. In agreement with ChIP data of GFP-Apj1*,

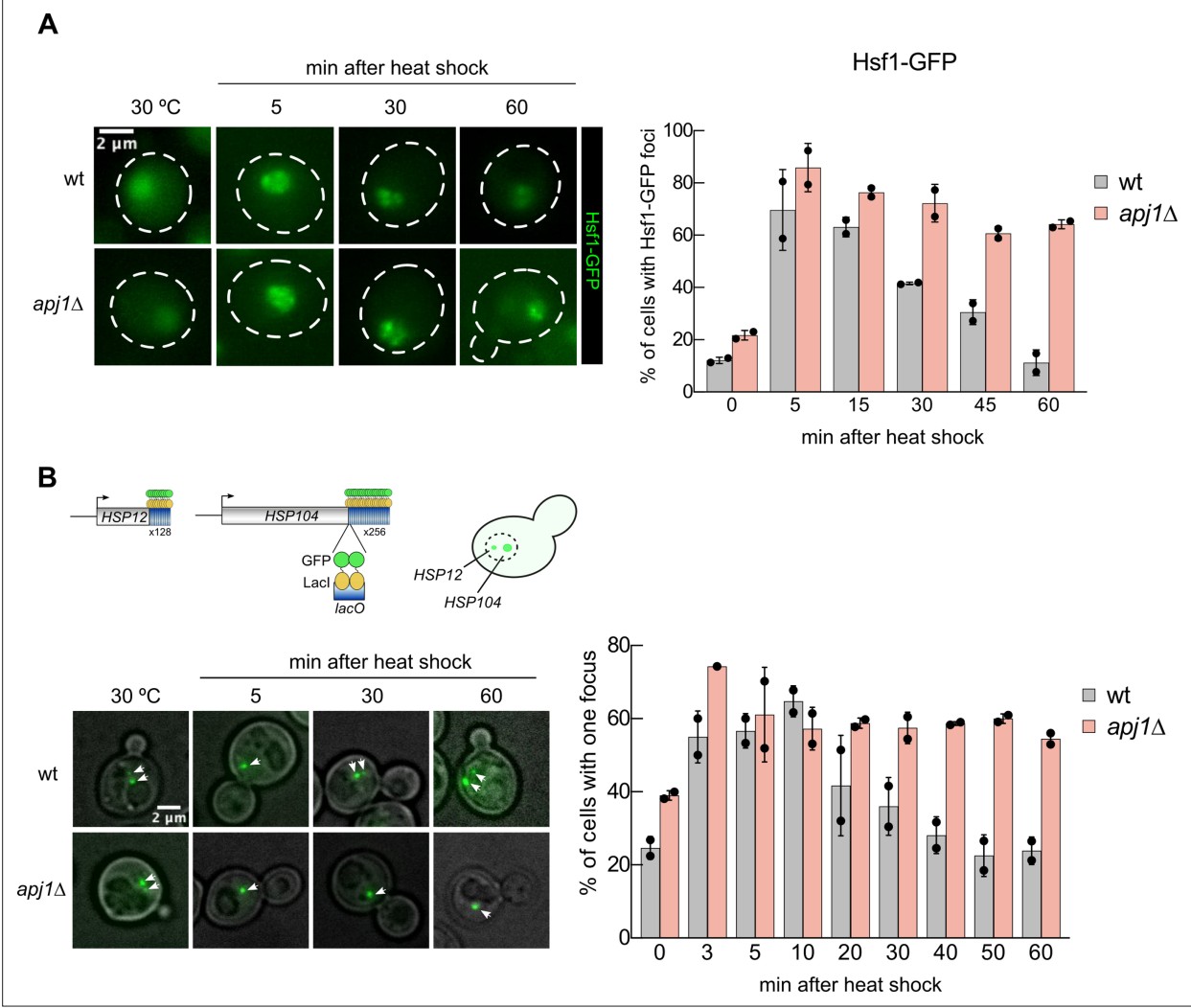

**Figure 2.** Persistent activation of Hsf1 in *apj1Δ* cells upon heat shock. (**A**) *S. cerevisiae* wt and *apj1Δ* cells expressing Hsf1-GFP were grown at 30°C and heat shocked to 38°C. Cellular localizations of Hsf1-GFP were determined at indicated time points and the proportions of cells showing two or more nuclear Hsf1-GFP foci were determined (n>221 for wt, n>133 for *apj1Δ*). (**B**) *S. cerevisiae* wt and *apj1Δ* cells expressing LacI-GFP and harboring *HSP12* and *HSP104* gene loci linked to 128 and 256 repeats of the *lacO* operator sequence, respectively, were grown at 30°C and heat shocked to 38°C. The percentage cells showing one or two LacI-GFP foci, reporting on coalescence of HSP104 and HSP12 gene loci, were determined at indicated time points (n>82 for wt, n>35 for *apj1Δ*). Scale bars are 2 µm.

Apj1-Myc$_{13}$ also bound specifically to UASs of heat shock genes (see *Figure 3—figure supplement 1C* for non-heat shock gene control *ARS504*), however, with maximum occupancies that were less pronounced, yielding a maximum of 3–4% of input controls as compared to 20–50% in case of Hsf1 (*Figure 3D*, *Figure 3—figure supplement 1B*), and at later time points during heat shock (20–60 min), showing an inverse relationship between Hsf1 and Apj1 occupancies. Validation with a synthetic *BUD3* gene fused to the *HSP82* UAS (*BUD3-HSE*) confirmed the anti-correlation (*Figure 3E*).

The observed anti-correlation suggested that Apj1 is necessary for dissociating Hsf1 from HSEs. Comparing Hsf1 occupancies at UASs of heat shock genes in wt and *apj1Δ* cells during heat shock revealed that Hsf1 occupancy did not decline in *apj1Δ* cells (*Figure 4*, *Figure 4—figure supplement 1A*), supporting a role for Apj1 in displacing Hsf1 from HSEs. Notably, Hsf1 occupancy was typically reduced at the early heat shock time points in *apj1Δ* cells compared to wt, although it increased with prolonged heat shock incubation. This was not observed for the *ARS504* control, documenting specificity for Hsf1 binding to HSEs (*Figure 4—figure supplement 1B*).

The significant defect of *apj1Δ* cells in repressing Hsf1 activity during the attenuation phase suggests that other JDPs, Ydj1 and Sis1, cannot compensate for Apj1. We assessed Sis1-Myc$_{13}$ and

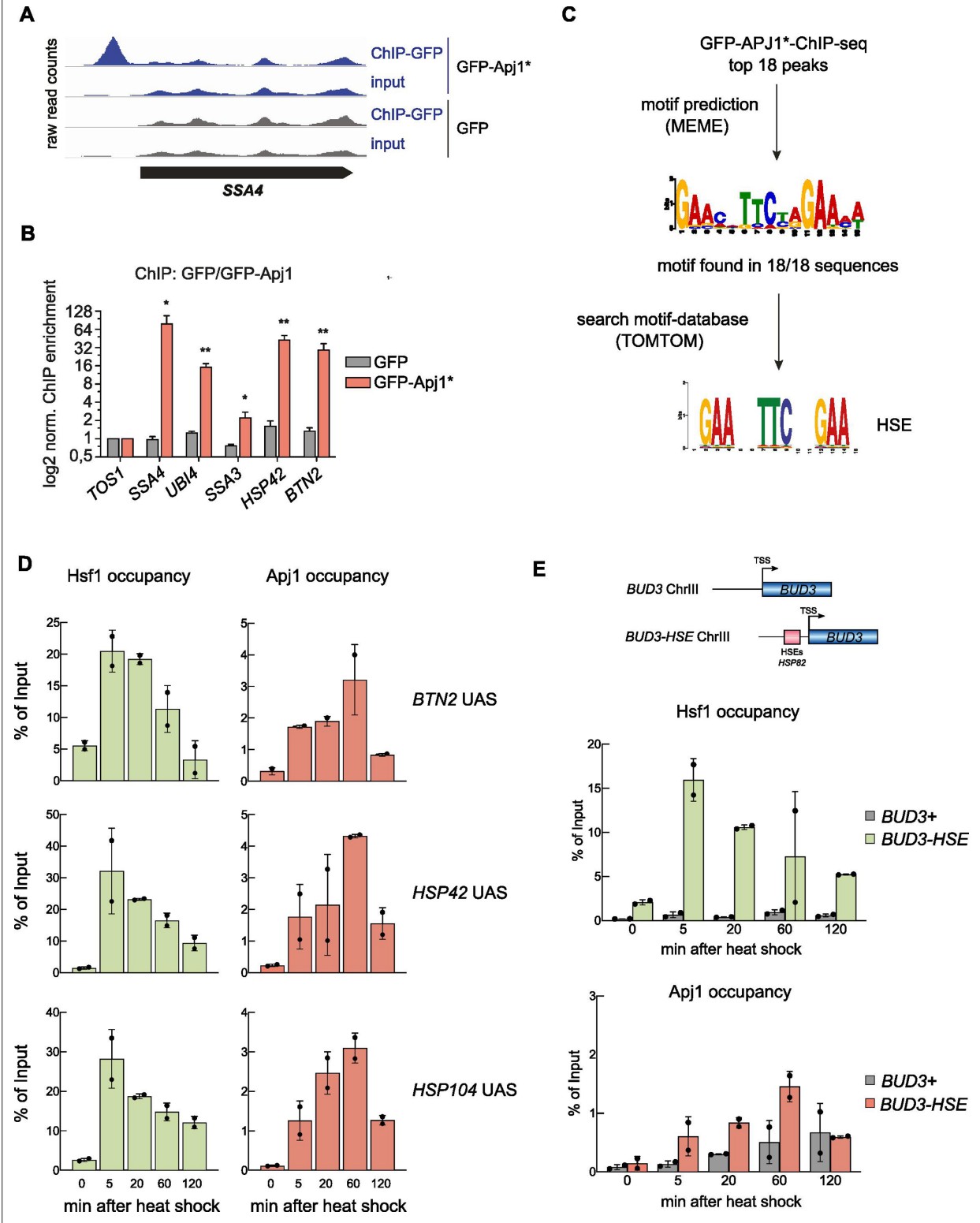

**Figure 3.** Chromatin occupancy of Hsf1 and Apj1 at *Heat Shock Response* genes is anti-correlated. (**A**) ChIP experiments using *S. cerevisiae* cells grown at 30°C and expressing GFP or GFP-Apj1* (GFP-Apj1-³⁴AAA³⁷). Cells were shifted to 42°C for 30 min, crosslinked, and processed for ChIP. Raw read counts of inputs and IPs for the heat shock gene *SSA4* (*HSP70*) are depicted. (**B**) ChIP enrichment (vs *TOS1* as control) of GFP-Apj1* and GFP for the indicated Hsf1 targets and the control *TOS1* are shown (n=3). Significance was determined by unpaired two-tailed *t*-test (*p<0.05; **p<0.01). (**C**) Top 18 binding peaks of GFP-Apj1* ChIP experiments were subjected to sequence analysis using MEME and searched for binding sites of transcription

*Figure 3 continued on next page*

*Figure 3 continued*

factors (using TOMTOM), revealing the Hsf1 target HSE (heat shock element). (**D**) Hsf1 and Apj1 occupancies at *UAS* regions of Hsf1-dependent heat shock gene loci. Occupancies were determined at the indicated time points after heat shock (30–39°C) by a ChIP assay. The percentage of input was calculated, and the mean values were plotted with SD (n=2). Statistical significance was determined relative to T=0 min sample by an unpaired two-tailed *t*-test. ns, p>0.05; *p<0.05; **p<0.01; ***p<0.001. (**E**) Hsf1 and Apj1 occupancies of *UAS* regions of the native *BUD3* gene and of *BUD3* that was placed under Hsf1 control (*BUD3-HSE*). The latter was accomplished by fusion of the *HSP82* promoter (containing three HSEs) with the minimal promoter region of *BUD3* (*Chowdhary et al., 2019*). Occupancies were determined at the indicated time points after heat shock (30–39°C) as above. The percentage of input was calculated, and the mean values were plotted with SD (n=2).

The online version of this article includes the following source data and figure supplement(s) for figure 3:

**Figure supplement 1.** Apj1 promotes the displacement of DNA-bound Hsf1 from HSE.

**Figure supplement 1—source data 1.** Original western blots with bands shown in *Figure 3—figure supplement 1A* highlighted.

**Figure supplement 1—source data 2.** Original western blots of *Figure 3—figure supplement 1A*.

Ydj1-Myc$_{13}$ occupancies at UASs in wt and *apj1Δ* cells, finding very low Ydj1 binding (<0.1%) with no significant changes during heat shock (*Figure 4—figure supplement 2A*). Similarly, we observed low Sis1 binding with a slight increase 20 min post heat stress at some UASs (*HSP104, HSP82*) (*Figure 4—figure supplement 2B*). Sis1 occupancy did not rise in *apj1Δ* cells (*Figure 4—figure supplement 2B*). We conclude that Apj1, rather than Ydj1 or Sis1, detectably interacts with the UAS regions of heat shock genes during heat stress.

Interplay of Apj1 with major JDPs Sis1 and Ydj1 in Hsf1 regulation *apj1Δ* cells are deficient in turning off the heat shock response but show no significant increase in Hsf1 activity in non-stressed conditions, as indicated, for example, by low Btn2 levels prior to heat shock (*Figure 1A*). This suggests compensatory activities of other JDPs maintaining Hsf1 in an inactive, non-DNA bound state prior to stress. To test for overlapping roles of JDPs in regulating Hsf1 in non-stressed cells, we linked the *APJ1* gene deletion to Ydj1 (*ydj1-4xcga*) and Sis1 (*sis1-4xcga*) knockdown cells in which depletion of Ydj1 and Sis1 was achieved through RQC (*Figure 5—figure supplement 1A*). We used *ydj1-4xcga* instead of *ydj1Δ* cells due to their much improved growth behavior at non-heat shock conditions (25°C). The degree of Ydj1 depletion (to 13.6 ± 2.6% of wt levels) was similar to Sis1 (14.9 ± 3%). We could not generate cells with double depleted Ydj1 and Sis1 levels, indicating their crucial overlapping functions in PQC.

The impact of the various JDP mutant strains on Hsf1 regulation was assessed by translatome analysis in cells grown at 25°C, as overlapping functions of JDPs in downregulating Hsf1 should activate the HSR in non-stressed JDP mutant cells. *apj1Δ* cells showed increased abundance of only a few Hsf1 target mRNAs (*SSA4, HSP42, HSP30*), while most targets did not exceed the two-fold increase threshold compared to wt (*Figure 5A*, *Figure 5—figure supplement 1B*). This supports the idea that Hsf1 remains largely repressed in non-stressed cells without Apj1. Similarly, Hsf1 activity barely increased in *ydj1-4xcga* cells, while partial HSR activation occurred in Sis1-depleted *sis1-4xcga* cells, consistent with previous findings (*Feder et al., 2021*; *Schilke and Craig, 2022*). The increase in translation of Hsf1 targets in *sis1-4xcga* cells varied, being most pronounced for *HSP104* and *SSA4* (*Figure 5A*, *Figure 5—figure supplement 1B*). Notably, the abundance of translated HSP mRNAs did not further increase in *apj1Δ sis1-4xcga* cells, indicating that another JDP maintains Hsf1 in a partially repressed state. This function might be fulfilled by Ydj1, as translatome analysis of *apj1Δ ydj1-4xcga* cells showed strong Hsf1 activation. Almost all Hsf1 targets exhibited increased abundance of HSP mRNAs in these mutants compared to wt, with greater upregulation than in all other JDP mutant strains tested (*Figure 5A*, *Figure 5—figure supplement 1B*). These results suggest dual control of Hsf1 activity by Ydj1 and Apj1, where loss of both JDPs ultimately triggers Hsf1 activation. The level of Hsf1 activation in *apj1Δ ydj1-4xcga* cells at 25°C resembled that of heat-shocked wt cells (10 min), although abundance of most Hsf1 targets increased further in heat shock samples ($R^2$=0.58) (*Figure 5—figure supplement 1C*), likely due to further depletion of Ydj1 through accumulation of misfolded proteins as well as heat shock-induced signaling that activates Hsf1 (*Zheng et al., 2016*). We additionally determined the transcriptome of WT and JDP mutant cells at 25°C and observed very similar consequences on Hsf1 activity (*Figure 5—figure supplement 2A/B*). Thus, *sis1-4xcga* cells showed partial and target-specific HSR activation, and strongest activation was again determined for *apj1Δ ydj1-4xcga* cells. These findings confirm that changes in Hsf1 activity can be faithfully monitored by translatome analysis.

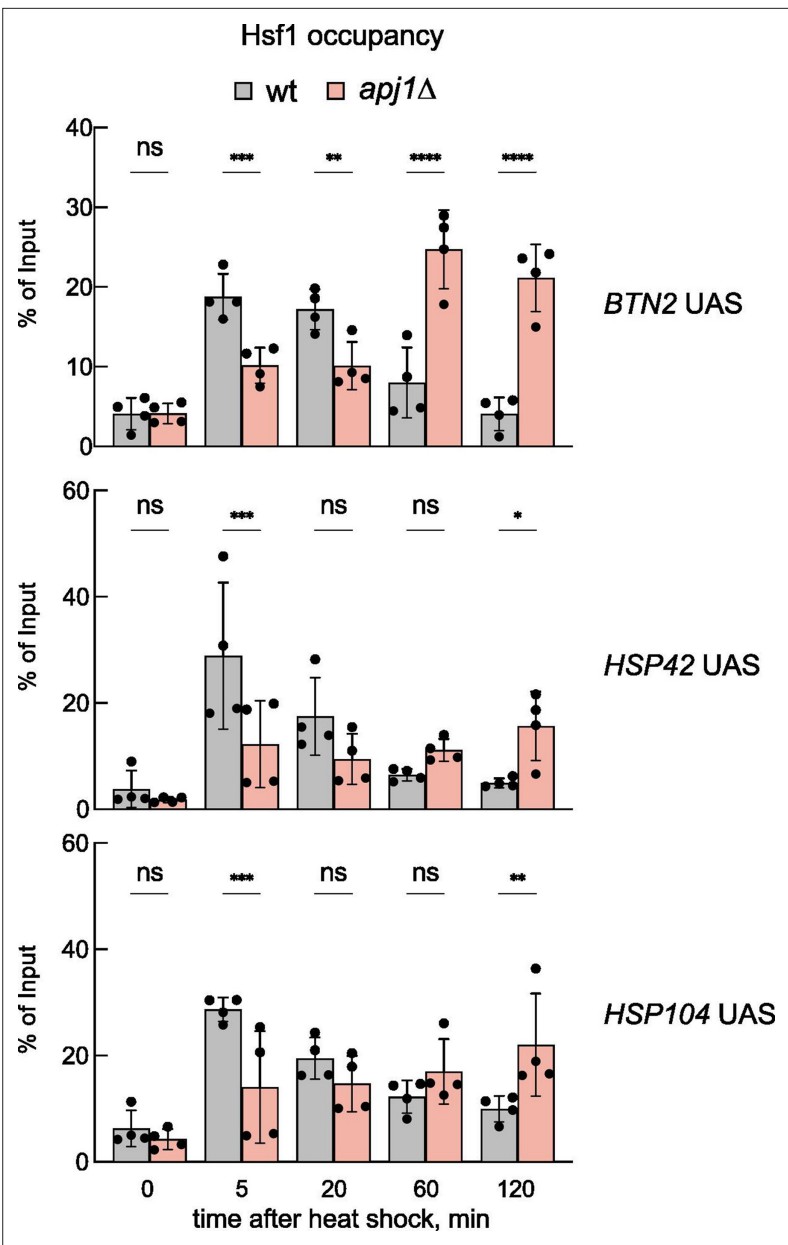

**Figure 4.** Hsf1 binding to *UAS* regions of heat shock genes is prolonged in *apj1Δ* cells. Hsf1 occupancies were determined in *S. cerevisiae* wt and *apj1Δ* cells at the indicated time points after heat shock (30–38°C) by a ChIP assay. The percentage of input was calculated, and the mean values were plotted with SD (n=4). Statistical significance (relative to T=0 min sample) was determined by one-way ANOVA: ns, p>0.05; *p<0.05; **p<0.01; ***p<0.001.

The online version of this article includes the following figure supplement(s) for figure 4:

**Figure supplement 1.** Hsf1 binding to *UAS* regions of heat shock genes is prolonged in *apj1Δ* cells.

**Figure supplement 2.** Apj1 occupancies at UAS of Hsf1-dependent heat shock genes are specific.

To confirm that HSR activation in *apj1Δ ydj1-4xcga* cells was not influenced by activated RQC, we generated *apj1Δ ydj1Δ* double knockout cells. These mutants exhibited a strong increase in protein levels of Hsf1 targets Btn2 and Hsp42, which hardly increased upon heat shock, indicating nearly full HSR activation under non-stress conditions (***Figure 5B***). We observed partially elevated Apj1 levels in *ydj1Δ* cells, which may prevent Hsf1 activation in the absence of Ydj1 (***Figure 5B***). We also found nearly complete Hsf1 activation in *apj1Δ ydj1Δ* cells, as evidenced by intense nuclear Hsf1-GFP foci

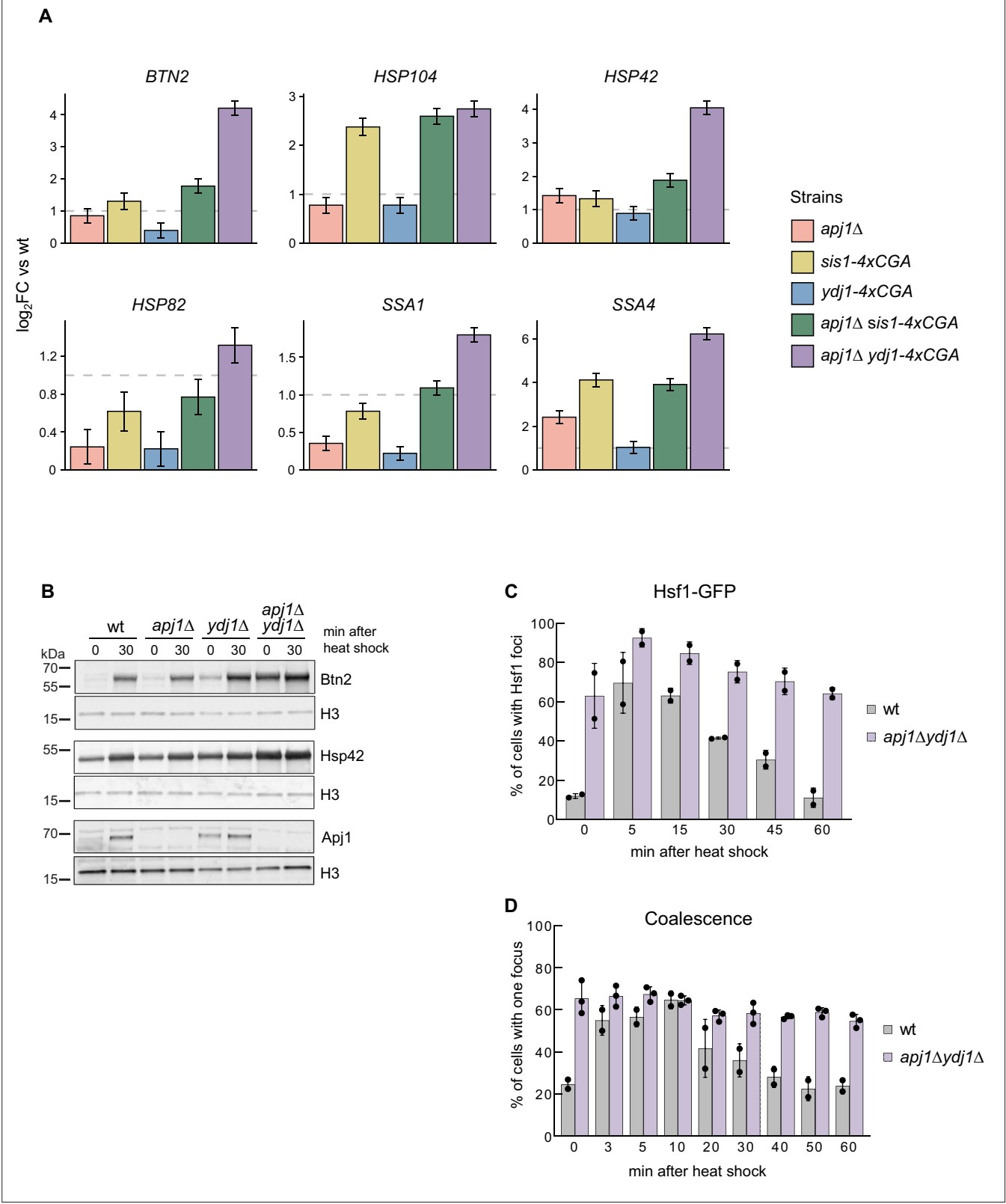

**Figure 5.** Loss of Apj1 and Ydj1 triggers almost full Hsf1 activation. (**A**) Changes in expression levels of selected Hsf1-target genes were determined by ribosome profiling in indicated yeast strains grown at 25°C. Levels of translated mRNAs in mutant strains were normalized to respective levels determined in wild-type cells and are shown in a log2-scale with the standard error (n=2–3). (**B**) Indicated yeast strains were grown at 25°C and then heat shocked at 35°C for 30 min. Total cell extracts were prepared and levels of the Hsf1 targets Btn2, Hsp42, and Apj1 were determined by western

*Figure 5 continued on next page*

**Figure 5 continued**

blot analysis. Levels of histone H3 are provided as loading control. (**C**) *S. cerevisiae* wt and *apj1Δ ydj1Δ* cells expressing Hsf1-GFP were grown at 30°C and heat shocked to 38°C. Cellular localizations of Hsf1-GFP were determined at indicated time points and the proportions of cells showing nuclear Hsf1-GFP foci were determined (n>221 for wt, n>203 for *apj1Δ*). (**D**) *S. cerevisiae* wt and *apj1Δ ydj1Δ* cells expressing LacI-GFP and harboring *HSP12* and *HSP104* gene loci linked to 128 and 256 repeats of the lacO operator sequence, respectively, were grown at 30°C and heat shocked to 38°C. The percentage of cells showing one or two LacI-GFP foci, reporting on coalescence of HSP104 and HSP12 gene loci, was determined at indicated time points (n>82 for wt, n>169 for *apj1Δydj1Δ*).

The online version of this article includes the following source data and figure supplement(s) for figure 5:

**Source data 1.** Original western blots with bands shown in *Figure 5B* highlighted.

**Figure supplement 1.** Loss of Apj1 and Ydj1 triggers Hsf1 activation.

**Figure supplement 1—source data 1.** Original western blots with bands shown in *Figure 5—figure supplement 1A* highlighted.

**Figure supplement 1—source data 2.** Original western blots of *Figure 1—figure supplement 1A*.

**Figure supplement 2.** Loss of Apj1 and Ydj1 triggers Hsf1 activation.

and coalescence of *HSP104* and *HSP12* genes in most mutant cells (63% and 65.5%, respectively) at 25°C (*Figure 5C/D*). These values showed little increase upon shifting to 35°C, demonstrating that loss of Ydj1 and Apj1 leads to full and persistent Hsf1 activation.

## Interplay of Hsf1 regulation via Apj1 and Ydj1 and cellular proteostasis

Next, we assessed how the various JDP mutants that affect Hsf1 activity impact proteostasis of yeast cells. We determined the temperature-sensitive growth phenotypes of the mutants through spot tests (*Figure 6A*, *Figure 6—figure supplement 1A*). Unlike *apj1Δ*, *sis1-4xcga* and *apj1Δ sis1-4xcga* cells, *ydj1-4xcga* and *ydj1Δ* cells showed impaired growth already at 30°C, in agreement with earlier findings (*Atencio and Yaffe, 1992*). Importantly, the growth defects of *YDJ1* single mutants were partially compensated in *apj1Δ ydj1-4xcga* or *apj1Δ ydj1Δ* double mutants, allowing growth up to 33–35°C (*Figure 6A*, *Figure 6—figure supplement 1A*). These findings indicate that strong Hsf1 induction enhances proteostasis in the double mutant cells by increasing PQC capacity. Notably, the improved proteostasis of *apj1Δ ydj1-4xcga* or *apj1Δ ydj1Δ* cells does not signal back to Hsf1 activity control as Hsf1 activity stays high, arguing in favor of direct roles of Ydj1 and Apj1 in controlling Hsf1.

To determine if enhanced Hsf1 activity rescues growth of *apj1Δ ydj1-4xcga* cells, we combined the JDP gene deletions with the *hsf1-848* allele, which reduces Hsf1 activity (*Yang et al., 2016*). The *apj1Δ ydj1-4xcga hsf1-848* cells did not grow at 30°C, confirming that HSR induction is essential for growth rescue (*Figure 6B*). Notably, the absence of Apj1 improved growth of *apj1Δ hsf1-848* cells at 37°C, consistent with a role of Apj1 in repressing Hsf1 function.

We next investigated whether Apj1's role as crucial regulator of Hsf1 activity involves cooperation with Hsp70, the established regulator of Hsf1 (*Krakowiak et al., 2018*; *Masser et al., 2019*; *Zheng et al., 2016*). We generated *ydj1-4xcga apj1-H34Q* cells to assess growth and chaperone levels (*Figure 6—figure supplement 1B*). The H34Q mutation in the Apj1 J-domain abrogates Hsp70 cooperation (*Rosenzweig et al., 2019*), leading to growth profiles comparable to *ydj1-4xcga apj1Δ* cells (*Figure 6—figure supplement 1B*). Consequently, *ydj1-4xcga apj1-H34Q* cells accumulated high Btn2 and Hsp42 levels under non-stress conditions (*Figure 6—figure supplement 1C*). Although Apj1-H34Q levels were also elevated, the mutant failed to downregulate Hsf1 activity, demonstrating that Apj1 acts as a canonical Hsp70 co-chaperone targeting Hsf1 to Hsp70 for repression.

To link the growth phenotypes of the JDP mutants to the respective proteostasis status, we assessed protein aggregation using Sis1-GFP as an aggregation reporter, as Sis1 binds and recruits Hsp70 to protein aggregates (*Figure 6—figure supplement 1D*; *Feder et al., 2021*; *Ho et al., 2019*; *Wyszkowski et al., 2021*). In wt cells, Sis1-GFP displayed diffuse fluorescence before heat shock and formed nuclear foci afterward, indicating Sis1-GFP recruitment to deposition sites of misfolded proteins (e.g., INQ) (*Ho et al., 2019*) and orphan ribosomal proteins (*Ali et al., 2023*; *Figure 6—figure supplement 1D*). This pattern is similar in *apj1Δ* cells, though more cells showed nuclear Sis1-GFP foci post-heat shock (53.7 vs 21% in *apj1Δ and wt* cells after 15 min, respectively, and 59.2 vs 30.7 after 60 min). In contrast, *ydj1Δ* cells showed multiple cytosolic and nuclear Sis1-GFP foci before heat shock (28.8% of cells) (*Figure 6—figure supplement 1D*). This number further increased upon heat shock and after 60 min massive cytosolic Sis1-GFP foci were noticed in 60.5% of *ydj1Δ* cells. This indicates

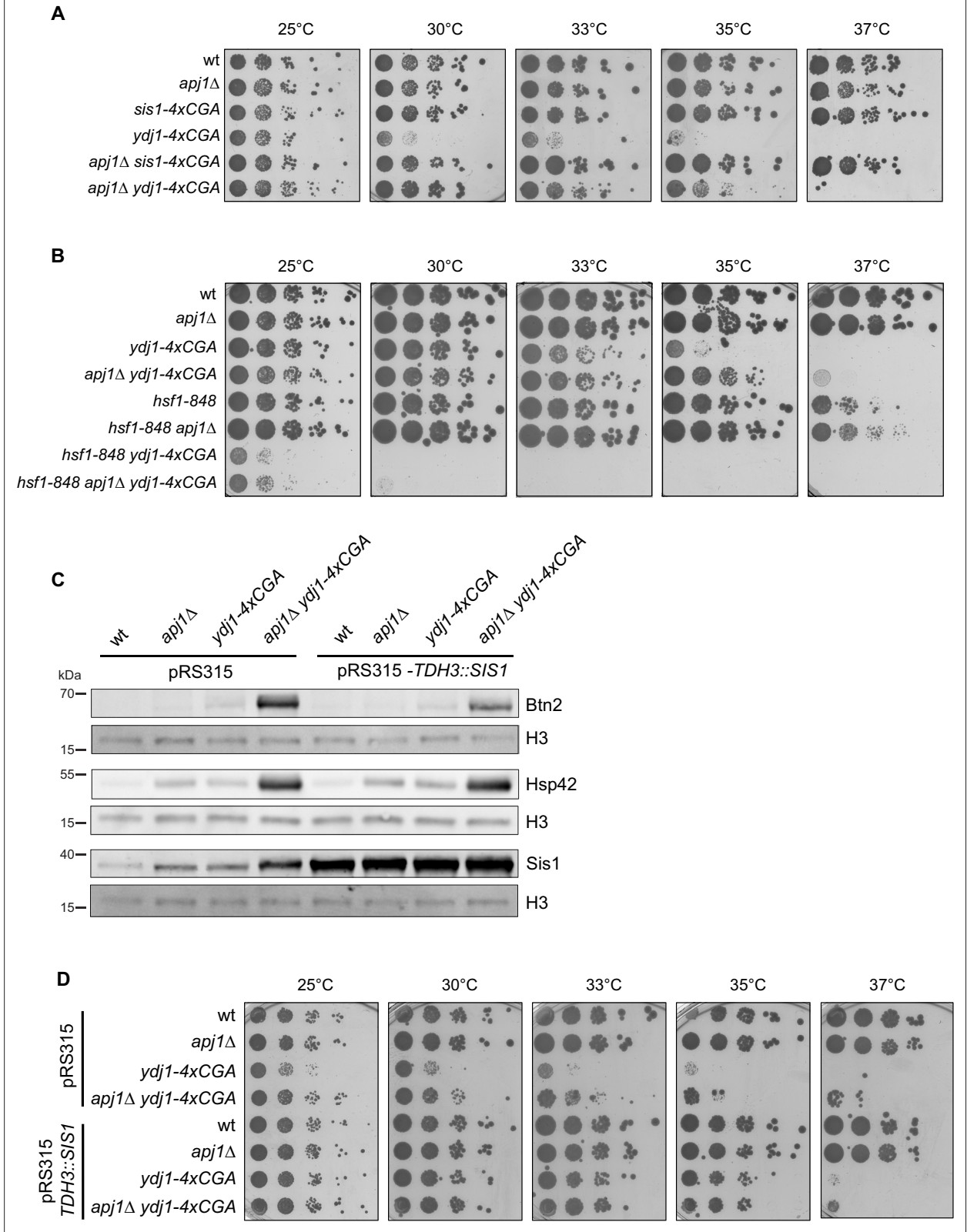

**Figure 6.** High Hsf1 activity rescues growth of *apj1 ydj1* mutant cells. (**A/B**) Serial dilutions of indicated yeast strains were spotted on YPD plates and incubated at indicated temperatures for 3 days. (**C**) *S. cerevisiae* wt, *apj1Δ, ydj1Δ,* and *apj1Δ ydj1Δ* cells overexpressing SIS1 (TDH3:Sis1) from plasmid or harboring an empty vector (EV) were grown at 25°C and levels of the Hsf1 targets Btn2 and Hsp42 were determined by western blot analysis. Levels of

*Figure 6 continued on next page*

*Figure 6 continued*

histone H3 were determined as loading control. (**D**) Serial dilutions of indicated yeast strains overexpressing Sis1 (TDH3::SIS1) from plasmid or harboring an empty vector (EV) were spotted on SC-Leu plates and incubated at indicated temperatures for 3 days.

The online version of this article includes the following source data and figure supplement(s) for figure 6:

**Source data 1.** Original western blots with bands shown in *Figure 6C* highlighted.

**Source data 2.** Original western blots of *Figure 6C*.

**Figure supplement 1.** Loss of Apj1 and Ydj1 triggers Hsf1 activation and restores proteostasis.

**Figure supplement 1—source data 1.** Original western blots with bands shown in *Figure 6—figure supplement 1C* highlighted.

**Figure supplement 1—source data 2.** Original western blots of *Figure 6—figure supplement 1C*.

---

severe proteostasis defects in *ydj1Δ* cells that agrees with the temperature-sensitive growth phenotype. An abnormal Sis1-GFP localization pattern was no longer observed in *apj1Δ ydj1Δ* cells and only nuclear Sis1-GFP foci were observed (*Figure 6—figure supplement 1D*). Notably, nuclear Sis1-GFP foci were already noticed prior to heat shock in 42.3% of *apj1Δ ydj1Δ* cells. This can be explained by (i) increased expression levels of the nuclear sequestrase Btn2 due to Hsf1 activation and (ii) absence of Apj1 increasing formation and stability of INQ. We also considered the possibility that formation of nuclear Sis1-GFP foci prior to heat shock in *apj1Δ ydj1Δ* cells liberates Hsf1, activating the HSR. However, the massive proteostasis defects of *ydj1Δ* cells and the resulting Sis1-GFP foci formation did not trigger Hsf1 activation. This argues against PQC defects as being the major reason triggering Hsf1 activation in *apj1Δ ydj1Δ* cells. It instead suggests that the growth deficiency of *ydj1Δ* cells arises primarily from Hsf1 repression via Apj1, limiting their PQC capacity adjustment.

We finally tested whether overproduction of plasmid-encoded Sis1 from the strong TDH3 promoter can restore Hsf1 activity control in *apj1Δ ydj1-4xcga* cells (*Figure 6C/D*). Strongly elevated Sis1 levels only partially reduced Btn2 levels while the high Hsp42 levels remained unaffected, indicating that Sis1 overproduction cannot efficiently repress Hsf1 activity (*Figure 6C*). We consider the possibility that a fraction of cells did not express SIS1 from the pRS315-based plasmid, which might lead to an underestimation of the effect. However, we also show that the increased Sis1 levels in *apj1Δ ydj1-4xcga* control cells harboring an empty vector are still insufficient to reduce Hsf1 activity, underlining that Sis1 cannot replace Apj1 and Ydj1 function in regulating Hsf1 (*Figure 6C*). Notably, Sis1 overproduction restored growth of *ydj1-4xcga* and *apj1Δ ydj1-4xcga* cells up to 35°C (*Figure 6D*), indicating overlapping functions of Sis1 and Ydj1 in yeast PQC and restoration of proteostasis. However, these beneficial effects of high Sis1 levels do not allow for Hsf1 repression, underlining the crucial and specific contribution of Apj1 to Hsf1 regulation.

## Discussion

In this study, we identified the nuclear JDP Apj1 as a crucial negative regulator of Hsf1 activity, specifically during the attenuation phase of the heat shock response (HSR). Apj1 functions by mediating the displacement of DNA-bound Hsf1 from HSEs, a role that is specific for Apj1 and cannot be replaced by other JDPs. Our findings suggest that different JDPs have specific roles in regulating distinct phases of the HSR: one set of JDPs maintains Hsf1 in a repressed state in non-stressed cells, while Apj1 represses Hsf1 following stress activation (*Figure 7*). Supporting this model, Sis1 has been implicated in repressing Hsf1 prior to stress but does not participate in HSR attenuation (*Feder et al., 2021*; *Garde et al., 2023*). In agreement, we observed partial derepression of Hsf1 upon Sis1 depletion, but no major effect on Hsf1 inactivation during the attenuation phase. That said, the impact of Sis1 depletion on the expression of Hsf1 targets in non-stressed cells is variable, ranging from minor (e.g. *BTN2*) to moderate (e.g., *HSP104*) derepression (*Figures 1C and 5A*). Differences in the number and spacing of HSEs within the UAS regions of the respective target genes are not obvious, and Hsf1 occupancies post-heat shock are comparable (*Pincus et al., 2018*). The molecular basis for the target-specific effects thus remains unclear.

The complementary roles of Sis1 and Apj1 in Hsf1 regulation predict a strong synergistic activation of Hsf1 upon Sis1 depletion in Apj1-deficient cells. Unexpectedly, however, this is not observed (*Figure 5*, *Figure 5—figure supplements 1 and 2*), indicating the involvement of additional JDPs in Hsf1 control under non-stress conditions. Our finding that simultaneous loss of Ydj1 and Apj1 resulted

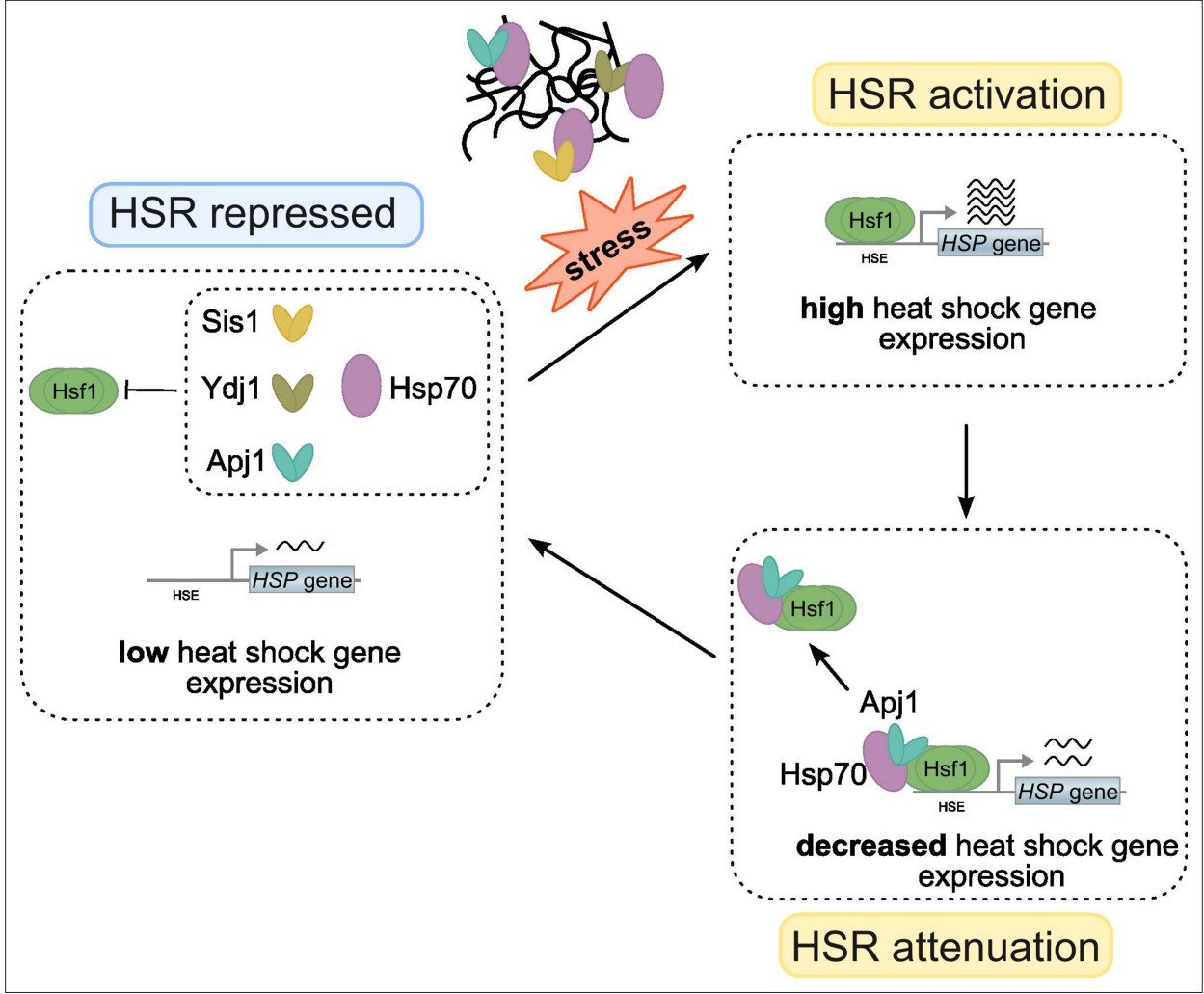

**Figure 7.** Regulation of Hsf1 activity via diverse J-domain proteins (JDPs). Hsp70 is targeted to Hsf1 in non-stressed cells by diverse J-domain proteins, including Ydj1, Sis1, and Apj1, to repress heat shock gene expression. Stress conditions trigger protein misfolding and aggregation. Binding of JDPs/Hsp70 to misfolded and aggregated proteins liberates Hsf1 to bind to heat shock elements (HSE) located upstream of heat shock genes, triggering their expression. Apj1 re-targets Hsp70 to HSE-bound Hsf1 during the attenuation phase, triggering Hsf1 dissociation and reducing heat shock gene expression. The HSR ultimately returns to the repressed state.

in almost full activation of Hsf1-target gene expression positions Yd1j as a central JDP that represses Hsf1 in non-stressed cells. How can JDPs specifically control different phases of the HSR? We speculate that this is due to distinct conformational states of free and HSE-bound Hsf1. Binding to HSEs may expose sites in Hsf1 specific for Apj1 targeting, while Sis1 and Ydj1 may target free Hsf1 conformations. Notably, two Hsp70 binding sites have been identified in Hsf1. The sites are located in the N- and C-terminal transactivation domains (*Krakowiak et al., 2018*; *Peffer et al., 2019*) and deletion of both sites leads to full Hsf1 activation (*Chowdhary et al., 2022*). It is tempting to speculate that the diverse JDPs specifically target Hsp70 to either N- or C-terminal binding sites.

Apj1 is present at low abundance (1000 copies/cell) in non-stressed cells compared to Sis1 and Ydj1 (28.000 and 50.000 copies/cell, respectively) (*Ho et al., 2018*). In accordance, our translatome analysis shows 22- and 46-fold higher translation of mRNA encoding SIS1 and YDJ1 as compared to APJ1 at 25°C (*Figure 1—figure supplement 2A*). Immediately after heat shock, the counts for all three mRNAs increase strongly, by 32-, 16-, and 4-fold for APJ1, SIS1, and YDJ1, respectively, underscoring all JDPs' roles in stress protection (*Figure 1—figure supplement 2B*). The low copy number of Apj1 may prevent excessive Hsf1 inactivation, maintaining basal activity crucial for adequate levels of essential chaperones like Hsp70 and Hsp90. Additionally, lower Apj1 levels may sensitize Hsf1

regulation in response to protein damage by allowing efficient depletion of Apj1 during the early phase of heat shock. Apj1 is involved in nuclear PQC, targeting misfolded and aggregated proteins for proteasomal degradation (*den Brave et al., 2020*). This function allows Apj1 to directly link the nuclear PQC status to Hsf1 activity regulation. The repair or removal of non-native proteins in the nucleus will free Apj1 to initiate the attenuation phase by targeting HSE-bound Hsf1. Consistent with this model, Apj1 binding to HSEs is delayed compared to Hsf1 (*Figure 3D/E*).

Both phases of the heat stress response, initiation and attenuation, are likely regulated by the titration of JDP cochaperones and their Hsp70 partners (*Figure 7*). It is remarkable that the initiation phase integrates protein misfolding information, for example, from newly synthesized proteins in the cytosol (*Masser et al., 2019*; *Tye and Churchman, 2021*) and orphan ribosomal proteins in the nucleolus (*Ali et al., 2023*; *Feder et al., 2021*), while the attenuation phase incorporates information on the general nuclear PQC status. A recent study demonstrated that disrupted protein repair in the cytosol (by mutating HSF1 target genes) delays HSR attenuation (*Garde et al., 2024*), which aligns with our model, as impaired cytosolic damage repair in those mutants could deplete nuclear Hsp70 resources needed for Hsf1 repression during attenuation. Additionally, misfolded proteins can be targeted to the nucleus for PQC (*Miller et al., 2015a*; *Park et al., 2013*; *Prasad et al., 2010*), which may also sensitize Hsf1 activity to accumulating protein damage. Apj1 also interacts with non-imported mitochondrial precursors (mitoPREs) (*den Brave et al., 2020*), which can be rerouted to the nucleus when mitochondrial import is blocked (*Shakya et al., 2021*). The accumulation of mitoPREs triggers a specific stress response involving Hsf1 (*Boos et al., 2019*), suggesting that Apj1 could act as a mediator, sensing mitoPRE presence and adjusting Hsf1 activity based on the status of mitochondrial import.

In summary, our study enhances the understanding of Hsf1 activity regulation by an array of JDPs and identifies Apj1 as an attenuation phase-specific regulator of the HSR. Future investigations should focus on the features of Apj1 that underlie its regulatory function. Although a direct homolog of Apj1 is absent in humans, the human class B JDP DnaJB1 can displace Hsf1 from HSEs in concert with Hsp70 in vitro (*Kmiecik et al., 2020*), hinting at a conserved function of JDPs in Hsf1 repression during the attenuation phase, albeit executed by different family members. Whether human class A JDPs can mimic Apj1's function in regulating Hsf1, and how diverse human JDPs cooperate to regulate the HSR needs to be explored.

## Materials and methods
### Yeast strains, growth conditions, and spot tests
All yeast strains used in this study are derived from BY4741 and W303 and are listed in *Supplementary file 1*. The plasmids used in this study are listed in *Supplementary file 2*. Yeast genome was edited using the homologous recombination technique as described (*Janke et al., 2004*). Desired genomic modifications were PCR amplified with flanking sequences homologous to the target genomic loci and suitable selection markers. PCR products were transformed in yeast cells and selected for the marker (auxotrophy or drug resistance). Correct integration at the locus was verified by PCR. Yeast cells were grown in liquid YPD or synthetic dropout (SC) media at indicated temperatures. SD medium was supplemented with 2% (w/v) glucose (Glu). Growth plates contained bacto-agar to a final volume of 2% (w/v). For spot tests, yeast cells were grown in SC or YPD medium to mid-log phase ($OD_{600}$=0.6–0.8) and then diluted to $OD_{600}$=0.2. Cells were fivefold serially diluted and spotted on YPD or SC agar plates. Plates were incubated at diverse temperatures for 2–3 days before documentation. Protein translation was stopped by addition of 100 µg/ml cycloheximide.

### Preparation of total cell lysates
Yeast cells were grown to 0.8 $OD_{600}$. 900 µl of cell culture was mixed with 150 µl of ice-cold 1.85 M NaOH, vortexed, and incubated on ice for 10 min. 150 µl of 55% (v/v) TCA was added and the suspension was gently mixed, followed by a further incubation on ice for 10 min. The mixture was centrifuged at 13,000 rpm, 4°C, 15 min. The supernatant was discarded, and the pellet was resuspended in 100 µl of HU buffer (8 M urea, 5% [w/v] SDS, 200 mM Tris pH 6.8, 1 mM EDTA, 1.5% [w/v] DDT, 0.1% [w/v]bromophenol blue) per 1 $OD_{600}$ equivalent. The resuspended pellet was incubated at 65°C for 10 min before analysis by SDS-PAGE.

## Western blotting

SDS-PAGEs were transferred to PVDF membranes by semi-dry blotting. Membranes were subsequently blocked with 3% BSA (w/v) in TBS-T. The antibodies and used dilutions are listed in *Supplementary file 3*. Anti-rabbit alkaline phosphatase conjugate (Vector Laboratories) was used as secondary antibody (1:20.000). Blots were developed using ECF Substrate (GE Healthcare) as reagent and imaged via Image-Reader LAS-4000 (Fujifilm). Western blotting was performed in two or more independent experiments each, and representative results are provided.

## Microscopy and image processing

For live-cell imaging, yeast cells were grown on SC medium up to OD 0.6 at 25°C, and a small volume (150 µl) was applied to Smart substrates (Interherence) previously treated with concanavalin A to immobilize the cells. Smart substrates were mounted on a VAHEAT temperature control unit (Interherence), and pictures were collected prior to and after heat stress application. To collect cell images, optical sections of 0.2 µm were acquired to image the whole-cell volume using a widefield system (Xcellence IX81, Olympus) equipped with a Plan-Apochromat 100× /NA 1.45 oil immersion objective and an EMCCD camera (Hamamatsu). Acquired z-stacks were deconvolved with xCellence software (Olympus) using the Wiener Filter. All further processing and analysis of digital images was performed with ImageJ (NIH).

## Chromatin immunoprecipitation (ChIP) and ChIP-seq

ChIP experiments (*Figure 3A/B*) were performed as described previously with minor modifications (*Aparicio et al., 2005*; *Kalocsay et al., 2009*). Briefly, cells were grown at 30°C to mid-log phase, subjected to heat shock at 42°C for 30 min and crosslinked with formaldehyde, chromatin was isolated and enriched for GFP or the indicated GFP-tagged proteins with a polyclonal GFP-antibody (ab290, Abcam). For next-generation sequencing, libraries of enriched and input DNA from duplicate or triplicate experiments were generated using the Microplex library preparation kit v2 (Diagenode) following the manufacturer's instructions and sequenced on a NextSeq 500 instrument (Illumina). Data processing was performed using the MACS software (*Zhang et al., 2008*) as described below. Data presented show raw read counts from one representative replicate.

For quantification by qPCR, duplicate measurements were performed and ChIP-DNA was normalized to input DNA for each tested locus and then normalized to a control region on chromosome II (*TOS1* promoter) that showed low fluctuations in ChIP-seq experiments for GFP. Hence, background levels were defined as 1.

For the ChIP-qPCR experiments shown in *Figures 3D and E and 4*, *Figure 3—figure supplement 1B/C*, *Figure 4—figure supplements 1 and 2*, cells from a 50 ml mid-log culture were crosslinked with 1% formaldehyde following heat shock at 39°C for the times indicated. 20% of chromatin lysates, generated as previously described (*Rubio et al., 2024*), were incubated with one of the following antibodies: 1.5 µl anti-Hsf1 antiserum or 2.5 µl anti-Myc monoclonal antibody (9E10; Santa Cruz Biotechnology) for 16 h at 4°C with gentle rotation. Antibody-bound chromatin was captured using Protein A or Protein G Sepharose beads (GE Healthcare, #17-09663-03 /#17-0618-01) by incubating for 16 h at 4°C. Chromatin was subsequently purified using the standard phenol-chloroform extraction method. Locus-specific primers were used to quantify ChIP DNA by quantitative PCR (qPCR) on a 7900HT Fast Real-Time PCR System (Applied Biosystems). Data was normalized to 25% input DNA, and the percentage of input was calculated accordingly (*Rubio et al., 2024*).

## RT-qPCR

For qPCR, total RNA was extracted using standard phenol-chloroform extraction method. The extracted RNA was treated with 1 U DNase I RNase free (Fermentas) for 30 min at 37°C. 1 µg of the resultant RNA was subjected to reverse transcription using Superscript III First Strand RT PCR Kit (Invitrogen) according to the instruction manual. Fresh cDNA was analyzed for various transcript levels, using DyNAmo Flash SYBR Green qPCR Kit (Thermo Scientific), transcript-specific primers, and Light Cycler 480 system (Roche). Transcript levels of all genes were normalized to that of actin. Sequences of RT-qPCR primers are provided in *Supplementary file 4*.

## GFP-Apj1-HPD* binding motif prediction

### Read processing and mapping

For all ChIP-Seq datasets, reads in FASTQ format were trimmed with the quality cutoff of 20 by fastq_quality_trimmer from FASTX Toolkit (version 0.0.14, http://hannonlab.cshl.edu/fastx_toolkit/). In addition, TrimGalore (version 0.4.5, https://github.com/FelixKrueger/TrimGalore; *Krueger and James, 2025*; *Martin, 2011*), which integrates Cutadapt, was used for adaptor trimming and reads shorter than 20 bp after adaptor and poly-A trimming were excluded before the mapping. Reads were mapped to *S. cerevisiae* S288C by applying segemehl (version 0.2.0) with 90% minimum alignment accuracy (*Hoffmann et al., 2009*). Gene quantifications were computed by using HTseq (version 0.10.0) (*Anders et al., 2015*). A principal component analysis was performed based on the information of gene quantification showing that input data and ChIP-Seq are significantly different.

### Peak calling and differential peak analysis

Model-based analysis of ChIP-Seq (MACS2, version 2.1.1) (*Zhang et al., 2008*) was applied to the ChIP-Seq and input datasets to detect significantly enriched ChIP regions for peak calling. DiffBind was used for detecting the significant up- and down-regulated genes when comparing the experimental data set and control (*Stark and Brown, 2011*). Differential peaks were visualized using the Integrative Genome Viewer software (v2.4.3). For each of the examples presented, the read count data range was adjusted to the highest peak from all tracks in the field of view, and the same data range was used for all tracks.

### Motif prediction

Based on the results of DiffBind, the sequences from 500 nts upstream of the differentially expressed genes were retrieved and used for predicting potential binding motifs. We selected the sequences with the strongest GFP-Apj1* enrichment (cut-off set to >13, 18 sequences) for further analysis. To predict potential binding sites in GFP-Apj1*-bound genomic regions, we used the MEME software suite version 5.5.8 (*Bailey et al., 2015*). We allowed zero to one motif or one motif occurrences per sequence and identified a motif present in 16 or 18 of the 18 sequences, respectively. Comparison of this motif with known transcription factor binding sites using the Tomtom motif comparison tool (version 5.5.8) (*Bailey et al., 2015*) revealed strong similarity with the known Hsf1 binding motif (HSE).

## Ribosome profiling sequencing library preparation

Ribosome profiling libraries were prepared as described in https://www.biorxiv.org/content/10.1101/2025.04.09.647970v1.full.

### Yeast culture, harvesting, and lysis for ribosome profiling

Overnight cultures of yeast cells were diluted in 500 ml YPD medium to an $OD_{600}$ of 0.05 and grown at 25°C or 30°C, shaking at 120 rpm. The cultures were harvested at $OD_{600}$ 0.4–0.5 by rapid filtration through a 0.45 um nitrocellulose membrane and flash frozen in liquid nitrogen.

The frozen cell pellets were lysed by mixer-milling with 600 µl lysis buffer pellets (20 mM HEPES-KOH pH 7.4, 150 mM KCl, 10 mM $MgCl_2$, 0.1 mg/ml cycloheximide, 0.1% [v/v] NP-40, 1x mini EDTA-free Roche protease inhibitor cocktail, 0.02 U/µl DNAse I, in DEPC-treated $H_2O$) in liquid nitrogen pre-cooled chambers, for 2 min at 30 Hz. The lysate was rapidly thawed in a 30°C water bath and clarified by centrifugation at 20.000 × *g* for 2 min at 4°C.

### RNA sequencing RNA isolation and fragmentation

100 µl phenol were extracted for total RNA. Poly-adenylated RNA was extracted using the New England Biolabs NEBNext Poly(A) mRNA Magnetic Isolation Module Express Protocol with a modified elution step. The hot supernatant was collected from the beads without prior cooling to 25°C. The resulting mRNA was fragmented with the NEBNext Magnesium RNA Fragmentation Module for 10 min and cleaned up using the Zymo RNA clean and concentrate kit and eluted in 6 µl RNAse free water.

## Footprint isolation

Lysate corresponding to 200 μg of RNA was digested with 50 U of RNAseI for 30 min on ice. The volume was brought to 400 μl with lysis buffer and loaded onto 800 μl sucrose cushion (30% [w/v] sucrose, 0.1 mg/ml cycloheximide, 20 mM HEPES-KOH [pH 7.5], 150 mM KCl, 10 mM MgCl$_2$, 1x EDTA free protease inhibitor cocktail). The cushion was centrifuged at 100.000 rpm, 4°C for 60 min. The supernatant was removed by aspiration, leaving only the 'lense' of ribosomes, which were resuspended in lysis buffer.

## Phenol extraction of RNA

100 μl of resuspended ribosomes was mixed with 400 μl Trizol (Zymo TRI) reagent. The mixture was agitated for 5 min at room temperature (RT), followed by a 3 min incubation. 200 μl chloroform was added, and the samples were vortexed for 30 s, then incubated for 5 min at RT. Phase separation was achieved by centrifugation at 20.000 × $g$ for 15 min. The upper aqueous phase (approximately 350 μl) was carefully transferred to a new tube. RNA was precipitated by adding 45 μl 3 M sodium acetate, 2 μl GlycoBlue (Invitrogen), and 450 μl isopropanol. Samples were stored at –80°C for at least 30 min to precipitate and centrifuged at 20.000 × $g$ for 45 min at 4°C. The pellet was washed with 0.75 ml ice-cold 70% ethanol and centrifuged at 20.000 × $g$ at 4°C for 5 min. After careful removal of residual ethanol, the pellet was dried at 55°C for 3 min and, if needed, dissolved in an adequate amount of 10 mM Tris (pH 7.0).

## Isolation of RNA footprints, recovery from gel, and precipitation

20 μg of RNA sample were mixed with 2x Novex TBE-Urea loading buffer and loaded on pre-run 15% TBE-Urea polyacrylamide gels for 65 min at 200 V. The gel was stained with SYBR gold, and the desired bands of 15–40 nt were cut out with a scalpel. The footprints were recovered from gel fragments by centrifugation at 30.000 × $g$ for 3 min in gel breaker tubes. The fragments were resuspended in 0.5 ml of 10 mM Tris (pH 7.0) and incubated at 70°C for 15 min with maximum agitation (1400 rpm) in a Thermomixer. The resulting gel slurry was transferred to a Spin-X cellulose acetate column and centrifuged at 20.000 × $g$ at 4°C for 3 min. The flow-through was collected in a non-stick tube and precipitated by adding 2 μl GlycoBlue, 55 μl 3 M sodium acetate (pH 5.5), and 0.5 ml isopropanol. Following 10 s of vortexing, samples were incubated at –80°C at least 30 min and centrifuged at 20.000 × $g$ for 45 min at 4°C. The pellet was washed with 0.75 ml ice-cold 70% ethanol and centrifuged at 20.000 × $g$ at 4°C for 5 min After careful removal of residual ethanol, the pellet was dried at 55°C for 3 min and used for library preparation.

## RNA dephosphorylation and linker ligation

The RNA pellet was resuspended in a reaction mixture containing 1×PNK buffer, murine RNase inhibitor, and 8 U polynucleotide kinase (PNK). Dephosphorylation was performed at 37°C for 1 h, followed by heat inactivation at 75°C for 10 min. For 3' linker ligation, a master mix containing 5 μM preadenylated biotinylated linker (App-N5-L1-biotin), 1×RNA ligase buffer, murine RNase inhibitor, 80U T4 RNA ligase II (truncated), and 25% (w/v) PEG 8000 was added to the reaction. Ligation was carried out at 22°C for 2 h. Excess linker was depleted by treatment with 25 U deadenylase and 45 U RecJF exonuclease at 37°C for 30–60 min, followed by heat inactivation at 75°C for 10 min.

## Bead binding, 5' end phosphorylation, and second linker ligation

Pierce MyOne streptavidin C1 beads were equilibrated in 1× binding and washing (BW) buffer (20 mM Tris pH 7.0, 500 mM NaCl, 1 mM EDTA, 0.05% Tween-20). Ligated RNA products were bound to the beads at RT for 10 min. The beads were washed twice with 1×BW buffer at 50°C and twice with 10 mM Tris (pH 7.0) containing 0.1% Tween-20 at 50°C. Each wash step included 5 min of incubation and magnetic separation.

Bead-bound RNA was phosphorylated using 4 U PNK in the presence of 1 mM ATP at 37°C for 1 h. 1 mM 5' L2 linker was ligated using 12U T4 RNA ligase I in the presence of 1 mM ATP and 25% (w/v) PEG 8000 at 37°C for 1 h. The beads were washed and resuspended in 10 mM Tris (pH 7.0).

## Reverse transcription and PCR amplification

Reverse transcription was performed using SuperScript III reverse transcriptase. The RT primer (FAM-RT, 100 µM) was annealed in the presence of dNTPs at 65°C for 5 min. First-strand synthesis was carried out at 50°C for 30 min. Following bead purification, PCR amplification was performed using Phusion Plus Polymerase (Thermo Scientific) with varying cycle numbers (5, 7, 9, and 11) to optimize amplification. PCR conditions were initial denaturation at 95°C for 2 min, followed by cycles of 95°C for 10 s, 64°C for 10 s, and 72°C for 10–20 s. PCR products were separated on a 6% TBE gel at 180 V for 40 min, with the target band migrating above 150 bp. The target band was cut out of the gel and recovered as described above. The sequencing was performed on an Illumina NextSeq 550 sequencer.

### Data analysis

The sequencing results were aligned to the genome as previously described (*Galmozzi et al., 2019*). In brief, the sequences were trimmed for adaptors and sequence/low-quality sequence using cutadapt version 3.5. Unique molecular identifiers (UMIs) were extracted from trimmed sequence reads using custom Julia script as described (*Bertolini et al., 2021*). Reads were aligned to *S. cerevisiae* rRNA sequences with bowtie version 1.3.1 using -n 2 alignment mode and `--best` reporting mode. Only reads that did not align were kept. Reads were mapped to *S. cerevisiae* S288C Genome (Assembly: GCA_000146045.2) using STAR version 2.7.10a. Following mapping parameters were used: `--outSAMmultNmax 1`; `--outFilterType BySJout`; `--alignIntronMin 5`; `--quantMode GeneCounts`; `--twopassMode Basic`; `--outSAMattributes` All XS. Reads from PCR duplicates were detected using UMI and were collapsed to a single read. Ribosomal positions were assigned.

The aligned counts of 2–3 independent biological replicates were analyzed using the DeSeq2 package in R. (*Love et al., 2014*). Genes with more than 20 reads were used in the analysis. The design of the DeSeq2 object includes the strain and the biological replicate, and the results were re-leveled to the wt. The samples for the timecourse experiment were treated as individual strains and the time variable was not included in the design formula. Translatome and transcriptome data are available via the NCBI Gene Expression Omnibus (Riboseq timecourse [*Figure 1C/D*]: GSE299855, Riboseq 25°C [*Figure 5A*]: GSE299862, RNAseq 25°C [*Figure 5—figure supplement 2*]: GSE298376). Analyzed translatome and transcriptome data are provided as *Supplementary files 5* (translatome timecourse data, *Figure 1C/D*), 6 (translatome data WT and JDP mutants, *Figure 5A*), and 7 (transcriptome data, *Figure 5—figure supplement 2*).

### AI usage statement

OpenAI's ChatGPT was used to assist in code development and troubleshooting. All conceptualization, data analysis, and final interpretations were conducted by the authors.

## Acknowledgements

This work was supported by a grant of the Deutsche Forschungsgemeinschaft to BB (BU617/21-1), GK (SPP2453, KR 3593/6-1, project ID 541620165), AM (SPP2453, MO970/9-1, project ID 541596792), and FdB (BR 6283/5-1, project ID 529716110; SPP 2453 BR 6283/6-1, project ID 541596792). It was also supported by funding from the U.S. National Institutes of Health to DSG (R01 GM138988). LS and AS were supported by the Heidelberg Biosciences International Graduate School (HBIGS).

## Additional information

### Funding

| Funder | Grant reference number | Author |
|---|---|---|
| Deutsche Forschungsgemeinschaft | BU617/21-1 | Bernd Bukau |
| Deutsche Forschungsgemeinschaft | MO970/9-1, project ID 541596792 | Axel Mogk |

| Funder | Grant reference number | Author |
|---|---|---|
| Deutsche Forschungsgemeinschaft | BR 6283/5-1, project ID 529716110 | Fabian den Brave |
| National Institutes of Health | R01 GM138988 | David S Gross |
| Deutsche Forschungsgemeinschaft | KR 3593/6-1, project ID 541620165 | Günter Kramer |

The funders had no role in study design, data collection and interpretation, or the decision to submit the work for publication.

## Author contributions

Carmen Ruger-Herreros, Lucia Svoboda, Conceptualization, Funding acquisition, Formal analysis, Investigation, Visualization, Methodology; Gurranna Male, Funding acquisition, Formal analysis, Investigation, Visualization, Methodology; Aseem Shrivastava, Formal analysis, Investigation, Visualization, Methodology; Markus Höpfler, Formal analysis, Investigation, Methodology; Katharina Jetzinger, Jiří Koubek, Funding acquisition, Formal analysis; Günter Kramer, Formal analysis, Writing – review and editing; Fabian den Brave, Conceptualization, Formal analysis, Funding acquisition, Visualization, Writing – review and editing; Axel Mogk, Conceptualization, Formal analysis, Supervision, Funding acquisition, Investigation, Visualization, Writing - original draft; David S Gross, Bernd Bukau, Conceptualization, Supervision, Funding acquisition, Investigation, Writing – review and editing

## Author ORCIDs

Carmen Ruger-Herreros ⬤ https://orcid.org/0000-0002-3982-9962
Lucia Svoboda ⬤ https://orcid.org/0009-0009-3035-2377
Gurranna Male ⬤ https://orcid.org/0000-0003-2544-6626
Aseem Shrivastava ⬤ https://orcid.org/0000-0002-3656-580X
Markus Höpfler ⬤ https://orcid.org/0000-0002-2129-2220
Katharina Jetzinger ⬤ https://orcid.org/0000-0003-0864-9448
Jiří Koubek ⬤ https://orcid.org/0000-0002-5117-6570
Günter Kramer ⬤ https://orcid.org/0000-0001-7552-8393
Fabian den Brave ⬤ https://orcid.org/0000-0001-6955-4616
Axel Mogk ⬤ https://orcid.org/0000-0003-3674-5410
David S Gross ⬤ https://orcid.org/0000-0002-7957-8790
Bernd Bukau ⬤ https://orcid.org/0000-0003-0521-7199

Reviewer #1 (Public review): https://doi.org/10.7554/eLife.107157.3.sa1
Reviewer #2 (Public review): https://doi.org/10.7554/eLife.107157.3.sa2
Reviewer #3 (Public review): https://doi.org/10.7554/eLife.107157.3.sa3
Author response https://doi.org/10.7554/eLife.107157.3.sa4

# Additional files

## Supplementary files

MDAR checklist

Supplementary file 1. Yeast strains used in this study.

Supplementary file 2. Plasmids used in this study.

Supplementary file 3. Antibodies used in this study.

Supplementary file 4. Sequences of RT-PCR primer used in this study.

Supplementary file 5. RiboSeq data showing time course of heat shock response.

Supplementary file 6. RiboSeq data of wt and JDP mutant strains at 25°C.

Supplementary file 7. RNAseq data of wt and mutant strains at 25°C.

## Data availability

All data are contained within the manuscript. Translatome and transcriptome data are available via the NCBI Gene Expression Omnibus (Riboseq timecourse (Figure 1C/D): GSE299855, Riboseq 25°C (Figure 5A): GSE299862, RNAseq 25°C (Figure 5 - Figure supplement 2): GSE298376). Analyzed translatome and transcriptome data are provided as *Supplementary file 5* (translatome timecourse data, Figure 1C/D), *Supplementary file 6* (translatome data WT and JDP mutants, Figure 5A) and *Supplementary file 7* (transcriptome data, Figure 5 - Figure Supplement 2).

The following datasets were generated:

| Author(s) | Year | Dataset title | Dataset URL | Database and Identifier |
| --- | --- | --- | --- | --- |
| Svoboda L, Male G, Shrivastava A, Höpfler M, Jetzinger K, Koubek J, Kramer G, den Brave F, Mogk A, Gross DS, Bukau B | 2025 | Nuclear and cytosolic J-domain proteins provide synergistic control of Hsf1 at distinct phases of the heat shock response [time course] | https://www.ncbi.nlm.nih.gov/geo/query/acc.cgi?acc=GSE299855 | NCBI Gene Expression Omnibus, GSE299855 |
| Svoboda L, Male G, Shrivastava A, Höpfler M, Jetzinger K, Koubek J, Kramer G, den Brave F, Mogk A, Gross DS, Bukau B | 2025 | Nuclear and cytosolic J-domain proteins provide synergistic control of Hsf1 at distinct phases of the heat shock response [steady state] | https://www.ncbi.nlm.nih.gov/geo/query/acc.cgi?acc=GSE299862 | NCBI Gene Expression Omnibus, GSE299862 |
| Ruger-Herreros C, Svoboda L, Male G, Shrivastava A, Höpfler M, Jetzinger K, Koubek J, Kramer G, den Brave F, Mogk A, Gross DS, Bukau B | 2025 | Nuclear and cytosolic J-domain proteins provide synergistic control of Hsf1 at distinct phases of the heat shock response | https://www.ncbi.nlm.nih.gov/geo/query/acc.cgi?acc=GSE298376 | NCBI Gene Expression Omnibus, GSE298376 |

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
