## [Editor Report · eLife Assessment]

This **valuable** study focuses on defining how the HSP70 chaperone system utilizes J-domain proteins to regulate the heat shock response-associated transcription factor HSF1. Using a combination of orthogonal techniques in yeast, this article provides **compelling** evidence that the J-domain protein Apj1 facilitates attenuation of HSF1 transcriptional activity through a mechanism involving its dissociation from heat shock gene promoter regions. This work generates new insight into the mechanism of HSF1 transcriptional regulation and is a significant contribution of broad interest to cell biologists interested in proteostasis, chaperone networks, and stress-responsive signaling.

---

## [Referee Report · Reviewer #1 (Public review)]

Summary:

In this study, the authors present a thorough mechanistic study of the J-domain protein Apj1 in *Saccharomyces cerevisiae*, establishing it as a key repressor of Hsf1 during the attenuation phase of the heat shock response (HSR). The authors integrate genetic, transcriptomic (ribosome profiling), biochemical (ChIP, Western), and imaging data to dissect how Apj1, Ydj1 and Sis1 modulate Hsf1 activity under stress and non-stress conditions. The work proposes a model where Apj1 specifically promotes displacement of Hsf1 from DNA-bound heat shock elements, linking nuclear PQC to transcriptional control.

Strengths:

Overall, the work is highly novel-this is the first detailed functional dissection of Apj1 in Hsf1 attenuation. It fills an important gap in our understanding of how Hsf1 activity is fine-tuned after stress induction, with implications for broader eukaryotic systems. I really appreciate the use of innovative techniques including ribosome profiling and time-resolved localization of proteins (and tagged loci) to probe Hsf1 mechanism. The overall proposed mechanism is compelling and clear-the discussion proposes a phased control model for Hsf1 by distinct JDPs, with Apj1 acting post-activation, while Sis1 and Ydj1 suppress basal activity.

The manuscript is well-written and will be exciting for the proteostasis field and beyond.

Comments on revised version:

The authors have addressed all my concerns,

---

## [Referee Report · Reviewer #2 (Public review)]

Summary:

Overall, the work is exceptionally well done and controlled and the results properly and appropriately interpreted. While several of the approaches, while powerful, are somewhat indirect (i.e., following gene expression via ribosomal profiling) additional experiments utilizing traditional gene expression assays added in revision combine to ultimately provide a compelling answer to the main questions being asked.

The key finding from this work is the discovery that Apj1 regulates Hsf1 attenuation in a manner that includes Hsp70. That finding is strongly supported by the experimental data. While it would be ideal to also demonstrate Apj1-controlled differential binding of Ssa1/2 to Hsf1 at either the N- or C-terminal binding sites during attenuation, the Hsp70-Hsf1 interactions are difficult to reproducibly assess in cell extracts and are likely beyond the scope of this study. However, this work paves the way in the future for potential biochemical reconstitution assays that could elucidate both Hsp70-Hsf1 interactions as well as the distinct JDP-Hsf1 interactions reported here.

This discovery raises additional new questions about JDP specificity in HSR regulation and the role of JDPs in navigating protein aggregation and sensing of proteostatic challenge in the nucleus, thus advancing the field and opening new, exciting avenues for exploration.

---

## [Referee Report · Reviewer #3 (Public review)]

Summary:

The heat shock response (HSR) is an inducible transcriptional program that has provided paradigmatic insight into how stress cues feed information into the control of gene expression. The recent elucidation that the chaperone Hsp70 controls the DNA binding activity of the central HSR transcription factor Hsf1 by direct binding has spurred the question how such a general chaperone obtains specificity. This study has addressed the next logical question, how J-domain proteins execute this task in budding yeast, the leading cell model for studying the HSR. While an involvement and in part overlapping function of general class A and B J-domain proteins, Ydj1 and Sis1 are indicated by the genetic analysis a highly specific role for the class A Apj1 in displacing Hsf1 from the promoters is found unveiling specificity in the system.

Strengths

The central strong point of the paper is the identification of class A J-domain protein Apj1 as a specific regulator of the attenuation of the HSR by removing Hsf1 from HSEs at the promoters. The genetic evidence and the ChIP data strongly support this claim. This identification of a specific role for a lowly expressed nuclear J-domain protein changes how the wiring of the HSR should be viewed. It also raises important questions regarding the model of chaperone titration, the concept that a chaperone with limiting availability is involved in a thug of war involving competing interactions with misfolded protein substrates and regulatory interactions with Hsf1. Perhaps Apj1 with its low levels and interactions with misfolded and aggregated proteins in the nucleus is the titrated Hsp70 (co)chaperone that determines the extent of the HSR? This would mean that Apj1 is at the nexus of the chaperone titration mechanism. Although Apj1 is not a highly conserved J domain protein among eukaryotes the strength of the study is that is provides a conceptual framework for what may be required for chaperone titration in other eukaryotes: One or more nuclear J-domain proteins with low nuclear levels that has an affinity for Hsf1 and that can become limiting due to interactions with misfolded Hsp70 proteins. The provides a pathway for how these may be identified using for example ChIP-seq.

Weakness

A built-in challenge when studying the mechanism of the HSR is the general role of Hsp70 chaperone system and its J domain proteins. Indeed, a weakness of the study is that it is unclear what of the phenotypic effects have to do with directly recruiting Hsp70 to Hsf1 dependent on a J domain protein and what instead is an indirect effect of protein misfolding caused by the mutation. This interpretation problem is clearly and appropriately dealt with in the manuscript text and in experiments but is of such fundamental nature that it cannot easily be fully ruled out.

---

## [Author Response]

The following is the authors’ response to the original reviews.

**Reviewer #1 (Public review):**

We thank the reviewer for his/her very positive comments.

**Reviewer #2 (Public review):**

We thank the reviewer for his/her positive evaluation. We plan to add RNAseq data of yeast wild-type and JDP mutant strains as more direct readout for the role of Apj1 in controlling Hsf1 activity. We agree with the reviewer that our study includes one major finding: the central role of Apj1 in controlling the attenuation phase of the heat shock response. In accordance with the reviewer we consider this finding highly relevant and interesting for a broad readership. We agree that additional studies are now necessary to mechanistically dissect how the diverse JDPs support Hsp70 in controlling Hsf1 activity. We believe that such analysis should be part of an independent study but we will indicate this aspect as part of an outlook in the discussion section of a revised manuscript.

**Reviewer #3 (Public review):**

We thank the reviewer for his/her suggestions. We agree that it is sometimes difficult to distinguish direct effects of JDP mutants on heat shock regulation from indirect ones, which can result from the accumulation of misfolded proteins that titrate Hsp70 capacity. We also agree that an in vitro reconstitution of Hsf1 displacement from DNA by Apj1/Hsp70 will be important, also to dissect Apj1 function mechanistically. We will add this point as outlook to the revised manuscript.

**Reviewer #1 (Recommendations for the authors):**
(1) Can the authors submit the raw translatome data to a standard repository? Also, the data should be summarized in a supplemental Excel table.

We submitted the raw translatome data to the NCBI Gene Expression Omnibus and added the analyzed data sets (shown in Figures 1 and 5) as Supplementary Tables S4/S5 (excel sheets). We additionally included RNAseq analysis of yeast WT and JDP mutants set grown at 25°C, complementing and confirming our former translatome analysis (new Figure 5, Figure Supplement 2). Respective transcriptome raw data were also deposited at the NCBI Gene Expression Omnibus and analyzed data are available as Supplementary Table S7.

(2) MW indicators need to be added to the Western Blot figures.

We added molecular weight markers to the Western Blot figures.

(3) Can the authors please include the sequences of the primers used in all the RT-qPCR experiments? They mention they are in the supplemental information, but I couldn't locate them.

We added the sequences of the RT-qPCR primers as Supplementary Table S4.

(4) Given the clear mechanism proposed, it would be nice if the authors could provide a nice summary figure.

We followed the suggestion of the reviewer and illustrate our main finding as new Figure 7.

**Reviewer #2 (Recommendations for the authors):**
(1) As mentioned above, a co-IP experiment between Hsf1 and Ssa1/2 in APJ1 and apj1∆ cells, utilizing Hsf1 alleles with and without the two known binding sites, would cement the assignment of Apj1 in the Hsf1 regulatory circuit.

We agree with the reviewer that Hsf1-Ssa1/2 pulldown experiments, as done by Pincus and colleagues (1), will further specify the role of Apj1 in targeting Hsp70 to Hsf1 during the attenuation phase of the heat shock response. We have tried extensively such pulldown experiments to document dissociation of Ssa1/2 from Hsf1 upon heat shock in yeast wild-type cells. While we could specifically detect Ssa1/2 upon Hsf-HA1 pulldown, our results after heat shock were highly variable and inconclusive and did not allow us to probe for a role of Apj1 or the two known Ssa1/2 binding sites in the phase-specific targeting. We now discuss the potential roles of the two distinct Ssa1/2 binding sites for phase-specific regulation of Hsf1 activity in the revised manuscript (page 12, lanes 17-21).

(2) Experiments in Figure 3 nicely localize CHIP reactions with known HSEs. A final confirmatory experiment utilizing a mutated HSE (another classic experiment in the field) would cement this finding and validate the motif and reporter-based analysis.

We thank the reviewer for this meaningful suggestions. We have done something like this by using the non-Hsf1 regulated gene BUD3, which lacks HSEs, as reference. We engineered a counterpart, termed “BUD3 HS-UAS”, which bears inserted HSEs, derived from the native UAS of HSP82, within the BUD3 UAS. We show that BUD3^+^ lacking HSEs is not occupied by Hsf1 and Apj1 under either non-stress or heat shock conditions while BUD3-HSE is clearly occupied under both, paralleling Hsf1 and Apj1 occupancy of HSP82 (Figure 3E). We have renamed the engineered allele to “BUD3-HSE” to clarify the experimental design and output.

(3) Page 8 - the ydj1-4xcga allele is introduced without explaining why it's needed, since ydj1∆ cells are viable. The authors should acknowledge the latter fact, then justify why the RQC depletion approach is preferred. Especially since the ydj1∆ mutant appears in Figure 5B.

ydj1∆ cells are viable, yet they grow extremely slowly at 25°C and hardly at 30°C, making them difficult to handle. The RQC-mediated depletion of Ydj1 in ydj1-4xcga cells allows for solid growth at 30°C, facilitating strain handling and analysis of Ydj1 function. Importantly, ydj1-4xcga cells are still temperature-sensitive and exhibit the same deregulation of the heat shock response upon combination with apj1D as observed for ydj1∆ cells. Thus ydj1 knockout and knockdown cells do not differ in the relevant phenotypes reported here and we performed most of the analysis with ydj1-4xcga cells due to their growth advantage. We added a respective explanation to the text (page 8, lanes 13-14) .

(4) The authors raise the possibility that Sis1, Apj1, and Ydj1 may all be competing for access to Ssa1/2 at different phases of the HSR, and that access may be dictated by conformational changes in Hsf1. Given that there are at least two known Hsp70 binding sites that have negative regulatory activity in Hsf1, the possibility that domain-specific association governs the different roles should be considered. It is also unclear how the JDPs are associating with Hsf1 differentially if all binding is through Ssa1/2.

We thank the reviewer for the comment and will add the possibility of specific roles of the identified Hsp70 binding sites in regulating Hsf1 activity at the different phases of the heat shock response to the discussion section. Binding of Ssa1/2 to substrates (including Hsf1) is dependent on J-domain proteins (JDPs), which differ in substrate specificity. It is tempting to speculate that the distinct JDPs recognize different sites in Hsf1 and are responsible for mediating the specific binding of Ssa1/2 to either N- or C-terminal sites in Hsf1. Thus, the specific binding of a JDP to Hsf1 might dictate the binding to Ssa1/2 to either binding site. We discuss this aspect in the revised manuscript (page 12, lanes 17-21).

(5) Figure 6 - temperature sensitivity of hsf1 and ydj1 mutants has been linked to defects in the cell wall integrity pathway rather than general proteostasis collapse. This is easily tested via plating on osmotically supportive media (i.e., 1M sorbitol) and should be done throughout Figure 6 to properly interpret the results.

Our data indicate proteostasis breakdown in ydj1 cells by showing strongly altered localization of Sis1-GFP, pointing to massive protein aggregation (Figure 6 – Figure Supplement 1D).

We followed the suggestion of the reviewer and performed spot tests in presence of 1 M sorbitol (see figure below). The presence of sorbitol is improving growth of ydj1-4xcga mutant cells at increased temperatures, in agreement with the remark of the reviewer. We, however, do not think that growth rescue by sorbitol is pointing to specific defects of the ydj1 mutant in cell wall integrity. Sorbitol functions as a chemical chaperone and has been shown to have protective effects on cellular proteostasis and to rescue phenotypes of diverse point mutants in yeast cells by facilitating folding of the respective mutant proteins and suppressing their aggregation (2-4). Thus sorbitol can broadly restore proteostasis, which can also explain its effects on growth of ydj1 mutants at increased temperatures. Therefore the readout of the spot test with sorbitol is not unambiguous and we therefore prefer not showing it in the manuscript.

**Author response image 1. sa4fig1:** Serial dilutions of indicated yeast strains were spotted on YPD plates without and with 1 M sorbitol and incubated at indicated temperatures for 2 days.

**Reviewer #3 (Recommendations for the authors):**
(1) Line 154: Can the authors, by analysis, offer an explanation for why HSR attenuation varies between genes for the sis1-4xcga strain? Is it, for example, a consequence of that a hypomorph and not a knock is used, a mRNA turnover issue, or that Hsf1 has different affinities for the HSEs in the promoters?

We used the sis1-4xcga knock-down strain because Sis1 is essential for yeast viability. The point raised by the reviewer is highly valid and we extensively thought about the diverse consequences of Sis1 depletion on levels of e.g. translated BTN2 (minor impact) and HSP104 (strong impact) mRNA. We meanwhile performed transcriptome analysis and confirmed the specific impact of Sis1 depletion on HSP104 mRNA levels, while BTN2 mRNA levels remained much less affected (new Figure 5 - Figure Supplement 2A/B). We compared numbers and spacings of HSEs in the respective target genes but could not identify obvious differences. Hsf1 occupancy within the UAS region of both BTN2 and HSP104 is very comparable at three different time points of a 39°C heat shock: 0, 5 and 120 min, arguing against different Hsf1 affinities to the respective HSEs (5). The molecular basis for the target-specific derepression upon Sis1 depletion thus remains to be explored. We added a respective comment to the revised version of the manuscript (page 12, lanes 3-8) .

(2) Line 194: The analysis of ChIP-seq is not very elaborated in its presentation. How specific is this interaction? Can it be ruled out by analysis that it is simply the highly expressed genes after the HS that lead to Apj1 appearing there? More generally: Can the data in the main figure be presented to give a more unbiased genome-wide view of the results?

We overall observed a low number of Apj1 binding events in the UAS of genes. The interaction of Apj1 with HSEs is specific as we do not observe Apj1 binding to the UAS of well-expressed non-heat shock genes. Similarly, Apj1 does not bind to ARS504 (Figure S3 – Figure Supplement 1). We extended the description of our ChIP-seq analysis procedures leading to the identification of HSEs as Apj1 target sites to make it easier to understand the data analysis. We additionally re-analysed the two Apj1 binding peaks that did not reveal an HSE in our original analysis. Using a modified setting we can identify a slightly degenerated HSE in the promoter region of the two genes (TMA10, RIE1) and changed Figure 3C accordingly. Notably, TMA10 is a known target gene of Hsf1. The expanded analysis is further documenting the specificity of the Apj1 binding peaks.

(3) Line 215. Figure 3. The clear anticorrelation is puzzling. Presumably, Apj1 binds Hsf1 as a substrate, and then a straight correlation is expected: When Hsf1 substrate levels decrease at the promoters, also Apj1 signal is predicted to decrease. What explanations could there be for this? Is it, for example, that Hsf1 is not always available as a substrate on every promoter, or is Apj1 tied up elsewhere in the cell/nucleus early after HS?

We propose that Apj1 binds HSE-bound Hsf1 only after clearance of nuclear inclusions, which form upon heat stress. Apj1 thereby couples the restoration of nuclear proteostasis to the attenuation of the heat shock response. This explains the delayed binding of Apj1 to HSEs (via Hsf1), while Hsf1 shows highest binding upon activation of the heat shock response (early timepoints). Notably, the binding efficiency of Hsf1 and Apj1 (% input) largely differ, as we determine strong binding of Hsf1 five min post heat shock (30-40% of input), whereas maximal 3-4% of the input is pulled down with Apj1 (60 min post heat shock) (Figure 3D). Even at this late timepoint 10-20% of the input is pulled down with Hsf1. The diverse kinetics and pulldown efficiencies suggest that Apj1 displaces Hsf1 from HSEs and accordingly Hsf1 stays bound to HSEs in apj1D cells (Figure 4). This activity of Apj1 explains the anti-correlation: increased targeting of Apj1 to HSE-bound Hsf1 will lower the absolute levels of HSE-bound Hsf1. What we observe in the ChIP experiment at the individual timepoints is a snapshot of this reaction. Accordingly, at the last timepoint (120 min after heat shock) analyzed, we observe low binding of both Hsf1 and Apj1 as the heat shock response has been shut down.

(4) Line 253: "Sis-depleted".

We have corrected the mistake.

(5) Line 332: Fig. 6C SIS1 OE from pRS315. A YIP would have been better, 20% of the cells will typically not express a protein with a CEN/ARS of the pRS-series so the Sis1 overexpression phenotype may be underestimated and this may impact on the interpretation.

We agree with the reviewer that Yeast Integrated Plasmids (YIP) represent the gold standard for complementation assays. We are not aware of a study showing that 20% of cells harboring pRS-plasmids do not express the encoded protein. The results shown in Fig. 8C/D demonstrate that even strong overproduction of Sis1 cannot restore Hsf1 activity control. This interpretation also will not be affected assuming that a certain percentage of these cells do not express Sis1. Nevertheless, we added a comment to the respective section pointing to the possibility that the Sis1 effect might be underestimated due to variations in Sis1 expression (page 11, lanes 15-19).

(6) Figure 1C. Since n=2, a more transparent way of showing the data is the individual data points. It is used elsewhere in the manuscript, and I recommend it.

We agree that showing individual data points can enhance transparency, particularly with small sample sizes. However, the log2 fold change (log2FC) values presented in Figure 1C and other figures derived from ribosome profiling and RNAseq experiments were generated using the DESeq2 package. This DeSeq2 pipeline is widely used in analyzing differential gene expression and known for its statistical robustness. It performs differential expression analysis based on a model that incorporates normalization, dispersion estimation, and shrinkage of fold changes. The pipeline automatically accounts for biological, technical variability, and batch effects, thereby improving the reliability of results. These log2FC values are not directly calculated from log-transformed normalized counts of individual samples but are instead estimated from a fitted model comparing group means. Therefore, the individual values of replicates in DESeq2 log2FC cannot be shown.

(7) Figure 1D. Please add the number of minutes on the X-axis. Figure legend: "Cycloheximide" is capitalized.

We revised the figure and figure legend as recommended.

(8) Several figure panels: Statistical tests and SD error bars for experiments performed in duplicates simply feel wrong for this reviewer. I do recognize that parts of the community are calculating, in essence, quasi-p-values using parametric methods for experiments with far too low sample numbers, but I recommend not doing so. In my opinion, better to show the two data points and interpret with caution.

We followed the advice of the reviewer and removed statistical tests for experiments based on duplicates.

References

(1) Krakowiak, J., Zheng, X., Patel, N., Feder, Z. A., Anandhakumar, J., Valerius, K. et al. (2018) Hsf1 and Hsp70 constitute a two-component feedback loop that regulates the yeast heat shock response eLife 7,

(2) Guiberson, N. G. L., Pineda, A., Abramov, D., Kharel, P., Carnazza, K. E., Wragg, R. T. et al. (2018) Mechanism-based rescue of Munc18-1 dysfunction in varied encephalopathies by chemical chaperones Nature communications 9, 3986

(3) Singh, L. R., Chen, X., Kozich, V., and Kruger, W. D. (2007) Chemical chaperone rescue of mutant human cystathionine beta-synthase Mol Genet Metab 91, 335-342

(4) Marathe, S., and Bose, T. (2024) Chemical chaperone - sorbitol corrects cohesion and translational defects in the Roberts mutant bioRxiv 10.1101/2024.09.04.6109452024.2009.2004.610945

(5) Pincus, D., Anandhakumar, J., Thiru, P., Guertin, M. J., Erkine, A. M., and Gross, D. S. (2018) Genetic and epigenetic determinants establish a continuum of Hsf1 occupancy and activity across the yeast genome Mol Biol Cell 29, 3168-3182